# USTBench: Benchmarking and Dissecting Spatiotemporal Reasoning Capabilities of LLMs as Urban Agents

**Siqi Lai, Yansong Ning, Zirui Yuan, Zhixi Chen, Hao Liu**[*]
AI Thrust, The Hong Kong University of Science and Technology (Guangzhou), Guangzhou, China
`{slai125,yning092,zyuan779}@connect.hkust-gz.edu.cn`
`zhixichen@hkust-gz.edu.cn, liuh@ust.hk`

## Abstract

Large language models (LLMs) have shown emerging potential in spatiotemporal reasoning, making them promising candidates for building urban agents that support diverse downstream applications. Despite these benefits, existing studies primarily focus on evaluating urban LLM agent on outcome-level metrics (*e.g.,* prediction accuracy, traffic efficiency), offering limited insight into their underlying reasoning processes. To this end, we introduce **USTBench**, the first benchmark to evaluate LLMs' spatiotemporal reasoning abilities as urban agents across four decomposed dimensions: *spatiotemporal understanding*, *forecasting*, *planning*, and *reflection*. Specifically, USTBench includes five urban decision-making and four spatiotemporal prediction tasks, all running within our interactive city environment *UAgentEnv*. The benchmark includes 62,466 structured QA pairs for process-based evaluation and standardized end-to-end task assessments, enabling fine-grained diagnostics and broad task-level comparison across diverse urban scenarios. Through extensive evaluation of fourteen leading LLMs, we reveal that although LLMs show promising potential across various urban downstream tasks, they still struggle in long-horizon planning and reflective adaptation in dynamic urban contexts. Notably, recent advanced reasoning models (*e.g.,* DeepSeek-R1) trained on general agent tasks do not consistently outperform non-reasoning LLMs. This discrepancy highlights the need for domain-specialized adaptation to enhance urban spatiotemporal reasoning. Overall, USTBench provides a foundation to build more adaptive and effective LLM-based urban agents and broad smart city applications. Our project is available at `https://github.com/usail-hkust/USTBench`.

## 1 Introduction

Urban systems are inherently complex and dynamic, characterized by continuous fluctuations across space and time. By learning from large-scale spatiotemporal data, traditional data-driven methods have achieved progress in prediction and decision support (Bibri & Krogstie, 2017; Ullah et al., 2020; Wei et al., 2019; Tian et al., 2025; Feng et al.). However, they often fall short in generalizing to unseen scenarios and providing transparent reasoning for reliable decision-making (Li et al., 2024c; Lai et al., 2025). Recently, the large language models (LLMs) (*e.g.,* o1 (Jaech et al., 2024) and DeepSeek-R1 (Guo et al., 2025)) have emerged as intelligent urban agents (Yuan et al., 2025; Li et al., 2024c; Zhou et al., 2024; Ning et al., 2025) due to their growing reasoning ability to integrate diverse information, adapt across tasks, and offer detailed interpretation through natural language. To fully leverage their potential, it is essential to systematically evaluate LLMs' spatiotemporal reasoning abilities: the capacity to infer spatiotemporal dynamics and interact with evolving urban environments. Such evaluation is key to understanding their readiness for real-world urban challenges.

In recent literature, many efforts have been made to evaluate the urban spatiotemporal reasoning ability of LLMs. However, as summarized in Table 1, they have two limitations: (1) *Reliance on outcome-based metrics*: solving urban tasks requires multi-step reasoning, yet existing studies (Feng et al., 2025a;b) only assess outcome-based metrics (*e.g.,* prediction accuracy, traffic efficiency), overlooking

---

[*]Corresponding author.

Table 1: Comparison of LLM benchmarks in urban tasks.

| Evaluations | | STBench (Li et al., 2024b) | CityBench (Feng et al., 2025b) | CityGPT (Feng et al., 2025a) | UrbanPlanBench (Zheng et al., 2025) | USTBench (Ours) |
|---|---|:---:|:---:|:---:|:---:|:---:|
| Reasoning Abilities | Spatiotemporal Understanding | ✓ | ✓ | ✓ | ✗ | ✓ |
| | Forecasting | ✓ | ✓ | ✓ | ✗ | ✓ |
| | Planning | ✗ | ✓ | ✓ | ✗ | ✓ |
| | Reflection | ✗ | ✗ | ✗ | ✗ | ✓ |
| Baseline LLMs | Non-Reasoning LLM | ✓ | ✓ | ✓ | ✗ | ✓ |
| | Reasoning LLM | ✗ | ✗ | ✗ | ✗ | ✓ |
| Evaluation Metrics | Outcome-Based Metrics | ✓ | ✓ | ✓ | ✓ | ✓ |
| | Process-Based Metrics | ✗ | ✗ | ✗ | ✗ | ✓ |

Figure 1: The comparison of outcome-based and process-based evaluations.

Figure 2: The performance of leading LLMs in urban spatiotemporal reasoning.

intermediate reasoning steps. Their evaluations may mask critical reasoning deficits. For instance, the reasoning LLM, DeepSeek-R1, typically surpasses non-reasoning models. However, as shown in Figure 1, it slightly underperforms Llama3.3 in outcome-based metrics of congestion prediction. Further reasoning process-based analysis reveals that this limitation stems from weaknesses in temporal trend understanding and forecasting. Without fine-grained evaluations, such discrepancies remain unexplained. (2) *Overlooking reflection reasoning*: unlike static tasks, urban systems provide real-time and context-rich feedback (*e.g.,* shifting traffic patterns), making reflection over past actions essential for agents to adapt to evolving dynamics (Wang et al., 2024c). Yet, existing evaluations fail to diagnose whether LLMs can transform this external feedback into causal links between prior actions and observed outcomes, and subsequently improve or adjust reasoning over time. These limitations prevent a comprehensive assessment of LLMs as urban agents.

To this end, we introduce *USTBench*, the first benchmark designed to systematically evaluate LLMs' spatiotemporal reasoning abilities as urban agents. We first build *UAgentEnv*, an interactive city environment spanning five typical decision-making and four prediction tasks. It enables agents to perceive, interact with, and respond to dynamic urban contexts. To move beyond outcome-based evaluation, we decompose urban spatiotemporal reasoning into four key processes: spatiotemporal understanding, forecasting, planning, and reflection. Each ability is assessed via question-answering (QA) pairs collected from UAgentEnv. To further explore the interplay between these reasoning processes, we conduct targeted ablation studies, providing diagnostic insights into model strengths and weaknesses. Combined with end-to-end performance evaluations, this dual-level framework supports both detailed reasoning analysis and standardized downstream task evaluation.

Using USTBench, we evaluate fourteen leading LLMs, covering both non-reasoning and reasoning models. Our key contributions and findings include: (1) We construct USTBench, the first benchmark explicitly designed to evaluate the spatiotemporal reasoning capabilities of LLMs as intelligent urban agents. It combines both process-based reasoning evaluation and standardized end-to-end task performance assessment, including 62,466 structured reasoning QA pairs and nine real-world downstream urban tasks. (2) To support this benchmark, we develop UAgentEnv, an interactive urban environment that enables nuanced benchmark dataset collection and uniform downstream task evaluation across diverse urban scenarios and tasks. (3) Despite the promising capabilities of LLMs, our extensive experiments also reveal several notable limitations: As illustrated in Figure 2,

LLMs excel in spatiotemporal understanding and forecasting, but typically struggles in long-term planning and reflection; Reasoning models trained on general logical or mathematical tasks do not consistently outperform non-reasoning LLMs, underscoring the need for domain adaptation; Planning is a higher-order ability that LLMs struggle with, which builds upon understanding and forecasting; Reflective learning, which enables adaptation through feedback in dynamic urban environments, remains a significant weakness, even among the leading reasoning models.

## 2 PRELIMINARY

**Problem 1. Urban Prediction Tasks**: Given historical spatiotemporal observations $\mathbf{o}_i$ of an urban environment, the goal of prediction is to anticipate urban future states $\{s_{i+1}, \ldots, s_{i+\Delta}\}$ over a horizon $\Delta$, where each state $s_i$ captures key indicators across space and time.

**Problem 2. Urban Decision-Making Tasks**: Given an urban environment and a task $t$, an agent operates based on observations $o$ with a policy $\pi(o)$ that determines a sequence of actions $\{a_0, a_1, ..., a_n\}$ to optimize the environment. The objective is to accomplish a target goal specified by $t$.

**Definition 1. Urban LLM Agents**: An urban LLM agent is a large language model-driven autonomous system designed to operate in dynamic urban environments. Formally, we define the urban environment as $E = \langle S, A, O, T \rangle$, where $S$ states the urban state space, $A$ denotes the agent's action space, $O$ is the observation space, and $T : S \times A \to S$ is the environment's transition function. At each time step $i$, given a task $t$, the agent receives the current observation $o_i \in O$ (*e.g.,* local traffic conditions), along with a history of prior observations $\mathbf{o}_{i-1}$ and actions $\mathbf{a}_{i-1}$. Based on this context, the agent performs reasoning to either (1) execute an action $a_i \in A$ (*e.g.,* activate a traffic signal), or (2) generate predictions of future urban states $\{s_{i+1}, \ldots, s_{i+\Delta}\} \subset S$ (*e.g.,* estimated traffic volume).

**Definition 2. Urban Spatiotemporal Reasoning**: Urban spatiotemporal reasoning is the agent's ability to interpret, act upon, and adapt to urban environments characterized by spatial and temporal dynamics. Formally, given a task $t$ and spatiotemporal observation $o_i$, it involves: (1) *Spatiotemporal Understanding* (Shi et al., 2022; Wang & Zhao, 2024): Interpreting urban spatial structures (*e.g.,* road network) and temporal patterns (*e.g.,* traffic flow shifts). (2) *Forecasting* (Wang et al., 2024c): Reasoning to generate predictions of future urban states $\{s_{i+1}, \ldots, s_{i+\Delta}\}$ based on learned spatiotemporal patterns. (3) *Planning* (Wang et al., 2024b): Reasoning to derive control actions $a_i$ that optimize task objectives within the current and anticipated urban context. (4) *Reflection* (Ji et al., 2023): Evaluating the outcomes of decisions or failures of predictions via external feedback $f_i$ and updating future reasoning accordingly to improve performance over time.

## 3 UAGENTENV: INTERACTIVE CITY ENVIRONMENT

### 3.1 URBAN TASK SUITE AND DATA SOURCES

**Urban Tasks**: UAgentEnv supports nine real-world urban tasks across prediction and decision-making. The prediction tasks include next POI prediction (Zhao et al., 2020), congestion forecasting (Cheng et al., 2018), socio-economic indicator prediction (Liu et al., 2023), and traffic origin-destination (OD) prediction (Yuan et al., 2024). The decision-making tasks cover traffic signal control (Wei et al., 2019), POI placement (von Wahl et al., 2022), route planning (Li et al., 2021), road planning (Zheng et al., 2023b), and urban planning (Zheng et al., 2023a). Details are in Appendix B.1.

**Urban Data Collection**: To ensure UAgentEnv comprehensively reflects urban dynamics and complexity, we integrate diverse publicly available real-world data across five dimensions: (1) *Geospatial data*: Geospatial information from OpenStreetMap (OSM) (OpenStreetMap contributors, 2025), including points of interest (POIs), areas of interest (AOIs), and road networks across multiple regions worldwide. (2) *Traffic*: Historical traffic flow data from several urban areas (Cheng et al., 2018; Zhang et al., 2023) to simulate realistic traffic conditions. (3) *Socio-economy*: Global GDP and population data from 2000 to 2019 (Wang & Sun, 2022; WorldPop, 2025) to model urban evolution. (4) *Human mobility*: City-wide trajectory data (New York City Taxi and Limousine Commission, 2025) capturing fine-grained movements of individuals within urban regions. (5) *POI check-ins*: Check-in records from the FourSquare (Wang et al., 2023) reflecting human mobility across POIs.

### 3.2 UNIFIED URBAN AGENT FRAMEWORK

We introduce a unified framework for urban tasks, where the urban agent interacts with the environment across nine real-world tasks. To ensure consistency, we standardize task instructions, inputs, outputs, and execution flows using simplistic instructions to showcase LLM basic reasoning abilities.

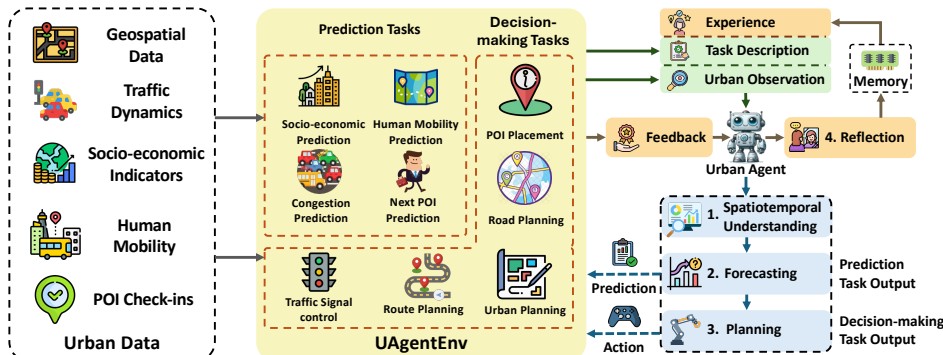

Figure 3: The workflow of UAgentEnv environment.

As shown in Figure 3, the framework follows a structured pipeline: (1) Each task provides the agent with a description, data schema, and relevant domain knowledge. (2) The real-time urban spatiotemporal dynamics are delivered to the agent as contextual observations. (3) The LLM agent reasons to solve the task through a modular reasoning workflow comprising spatiotemporal understanding, forecasting, and planning. Then, an action or prediction is generated, aligning with the task objective and relevant past experiences retrieved from memory. (4) After receiving environmental feedback on the previous output, the agent engages in reflection to evaluate its performance and diagnose errors. Then, an informative experience will be generated and stored in the memory to guide future reasoning processes. The prompt templates are detailed in Appendix B.3.

# 4 USTBench: Urban Spatiotemporal Reasoning Benchmark

## 4.1 Process-based Spatiotemporal Reasoning Evaluation

USTBench provides a fine-grained assessment by decomposing urban spatiotemporal reasoning into four key processes, which represent a loop of agent-environment interaction: (1) understanding the spatiotemporal dynamics (Shi et al., 2022; Cao et al., 2024), (2) forecasting the future states of the environment (Wang et al., 2024c), (3) planning actions based on anticipated changes(Chang et al., 2024), and (4) reflecting on environmental feedback to improve subsequent reasoning(Renze & Guven, 2024). These abilities are foundational skills widely recognized (Ni et al., 2024; Kim et al., 2025) for LLM agents to interact with dynamic environments effectively.

Existing benchmarks primarily focus on fundamental spatiotemporal understanding or outcome-based metrics. In contrast, USTBench offers process-based diagnostics that cover the full understand–forecast–plan–reflect loop, enabling systematic analysis of where reasoning succeeds or fails short. The benchmark consists of 62,466 structured QA pairs spanning nine representative urban tasks, including 40% for basic spatiotemporal understanding, and 60% for higher-level reasoning in real-world task-solving. These reasoning abilities are evaluated with accuracy, delivering a coherent, interpretable, and empirically grounded framework for assessing LLM agents' spatiotemporal reasoning. Statistics and examples are shown in Table 2, with details provided in Appendix D.7.3.

### 4.1.1 Observation Construction

To systematically evaluate LLMs' reasoning capabilities across diverse urban environments, we collect a variety of task scenarios as LLM observations using UAgentEnv. Each observation encodes a snapshot of geospatial structures (*e.g.,* road networks) and temporal dynamics (*e.g.,* trajectories or time-series) of the city. These serve as contextual input for LLMs to perform reasoning over specific urban questions. We verbalize spatial observations as sparse adjacency matrices with node and edge attributes (Chen et al., 2024). Temporal observations are verbalized with attribute values (*e.g.,* visited POI or traffic volume) across discrete time intervals (Wang et al., 2023; Li et al., 2024c). Examples of observations are shown in Table 2, and detailed prompts are in Appendix D.7.3.

To collect QAs from prediction tasks, we construct observations using a sliding window of length $\Delta$. At each step $i$, we extract real-world historical spatiotemporal dynamics and encode as input observations in QA queries. For QAs constructed from decision-making tasks, we collect observations using a heuristic agent that follows a semi-stochastic policy to interact with the environment for

Table 2: The required reasoning abilities of LLM agents and the statistics of our evaluation dataset.

| Reasoning | Category | Observation Example | Question Example |
|---|---|---|---|
| Spatiotemporal Understanding (27,000 QAs) | Distance | Road length: road1: 126m, road2: 345m... | Rank roads by their distances. |
| | Adjacency | Adjacency: [(region1, region2, 341m), (region1, region3, 125m),...] | Rank the regions by proximity to region 1. |
| | Connectivity | Connectivity: [(road1, road2, 121m), (road4, road5, 156m),...] | Is there a path between roads 1 and 5? |
| | Duration | Travel times: route 1: 34min, route 2: 12min,... | Rank the routes by their travel times. |
| | Chronology | Trajectory: [(Shop, 19:34),...,(Shop, 20:21), (Bar, 20:41),...] | Order the often visited POI pairs. |
| | Trend | Congestion levels: [(2, 8:00), (3, 8:05), (4, 8:10),...] | What is the trend of the congestion? |
| | Local Extrema | Congestion levels: [(4, 10:00),..., (3, 10:00),...,(4, 10:00),...] | Identify high-peak hours in the last 3 days. |
| | Periodicity | Congestion levels: [(4, 9:00),..., (2, 16:00),...,(4, 21:00),...] | Identify the period of the traffic flow pattern. |
| Forecasting (15,336 QAs) | Next POI | Trajectory: [(Shop, 19:34, 0m), (Bar, 20:11, 134m),...] | Predict the next POI the user would go. |
| | Congestion | Congestion levels: [(2, 8:00), (3, 8:05),...,(4, 8:55)] | Predict the congestion at 9:00. |
| | Socio-economic | Region GDP: [(63M, 2009), (68M, 2010), (71M, 2011),...] | Predict the GDP in 2012. |
| | Traffic-OD | Departure vehicles: [(5, 8:00), (10, 8:05),...,(6, 8:55)] | How many vehicles will depart at 9:00. |
| Planning (15,000 QAs) | Signal Control | Queues: lane1: 3, lane2: 9,...; Connectivity: [(lane1, lane3, 300m),...] | Which signal should be activated? |
| | POI Placement | Demand: loc1: 12, loc2: 23,...; Distance: loc1: 1021m, loc2: 2033m,... | Where should we placed a new station? |
| | Route Planning | Congestion level: road1: 1, road4: 2,...; Connectivity: [(road1, road2, 126m),...] | Which road should enter next? |
| | Road Planning | Connectivity: [(road1, region2, 134m), (road3, region3, 234m),...] | Which road should be built next? |
| | Urban Planning | Adjacency: [(blank1, residential1, 143m), (blank1, residential2, 345m),...] | Where should we plan a new hospital? |
| Reflection (8,130 QAs) | Reflection on Forecasting | Ground truth: (4, 10:00); Your prediction: (1, 10:00) | Your previous prediction is wrong, let's review the spatiotemporal data and try again. |
| | Reflection on Planning | The facility accessibility is decreased by 12%. | Verify if the previous decision and its reasoning are accurate. |

varying urban contexts with diverse spatiotemporal dynamics:

$$\pi_g(o) = \begin{cases} \arg\max_{a \in A} Q(o, a), & \text{with probability } 1 - \epsilon \\ \text{random}(A), & \text{with probability } \epsilon \end{cases} \quad (1)$$

where $Q(o, a)$ is a simple utility function that scores the benefit of taking action $a$ in state $s$ (*e.g.,* prioritizing the green signal for lanes with the longest queues). To ensure diversity of collected scenarios, we introduce an exploration coefficient $\epsilon \in [0, 1]$, which controls the probability of selecting a random action. This induces diverse decision trajectories resulting in various decision-making scenarios. At each decision time step $i$, the observed spatiotemporal dynamics are captured and embedded into a corresponding QA instance as agent observations.

### 4.1.2 SPATIOTEMPORAL UNDERSTANDING QA

Spatiotemporal understanding forms the first stage of problem-solving and underpins all interactions with urban environments. It requires LLMs to interpret city-scale spatial relations and identify dynamic temporal patterns within noisy urban environments. To evaluate this ability, we design QA tasks using contextual observations from Section 4.1.1, where each instance challenges the LLM to infer spatial structures or temporal patterns from the provided context. Our benchmark integrates eight widely recognized types of spatiotemporal patterns, following established definitions from prior works, enabling systematic assessment of whether LLMs can correctly extract these features.

**Spatial Understanding** extends from local neighborhood to long-range city-wide dependencies. (1) *Distance* reasoning captures the proximity between locations (Rizvi et al., 2024). (2) *Adjacency* reasoning extends to assessing whether multiple nearby locations form a coherent spatial neighborhood (Shi et al., 2022). (3) *Connectivity* requires multi-step inference across long-range urban topological relations to find whether a viable path exists between spatial entities (Zheng et al., 2023b;a).

**Temporal Understanding** progresses from localized analysis to long-horizon inference. (1) *Duration* reasoning focuses on comparing the lengths of individual events (Cao et al., 2024). (2) *Local Extrema* detection extends to identifying peak or off-peak periods within short temporal windows (Wang & Zhao, 2024). (3) *Chronology* reasoning then infers the temporal order and dependencies among multiple events over broader timescales (Xiong et al., 2024). (4) *Periodicity* reasoning integrates long-term observations to uncover recurring seasonal cycles in urban dynamics (Wang & Zhao, 2024). (5) *Trend* analysis identifies persistent directional changes over extended horizons, capturing long-term growth or decline patterns (Liu et al., 2025a).

### 4.1.3 FORECASTING QA

Building up spatiotemporal understanding, forecasting (Wang et al., 2024c; Li et al., 2024c; Tian et al., 2026) allows agents to predict future urban states. This is an essential capability not only for prediction but also for enhancing action-outcome modeling in downstream planning (Fu et al., 2025). To evaluate forecasting as a standalone ability, we construct QAs from the prediction tasks in

UAgentEnv. Based on the historical observations $\mathbf{o}_i$, the agent predicts the future urban state at the next timestep $s_{i+1}$. The ground truth is derived from the actual observed value in the real-world data.

### 4.1.4 PLANNING QA

Planning (Wang et al., 2024b; Chang et al., 2024) reflects the agent's ability to reason over spatiotemporal observations and choose actions that optimize long-term urban objectives. Unlike solving static problems (*e.g.,* mathematics, web search), planning in urban tasks requires agents to consider benefits over an extended horizon within the complex and evolving environments. To assess this ability, we construct QA instances from five urban decision-making tasks. Given the current spatiotemporal observation $o_i$, the agent is tasked to select an action $a_i \in A$ that best advances the task objective (*e.g.,* reducing traffic congestion). Because real-world urban systems rarely reveal optimal actions due to stochasticity and delayed feedback, we compute the ground-truth answer through a simulation-driven exhaustive search. Specifically, we evaluate all possible future action sequences over a planning horizon $H$ and select the action $a_i^*$ that maximizes the expected cumulative reward:

$$a_i^* = \arg\max_{a_i \in A} \max_{a_{i+1},\ldots,a_{i+H} \in A} \mathbb{E}\left[\sum_{j=0}^{H} \gamma^j R(a_{i+j}) \mid a_i\right], \tag{2}$$

where $R(a_{i+j})$ is the reward when action $a_{i+j}$ is taken. Here, the reward is the measure of progress toward the specific objective of the task (*e.g.,* the reduction in queue length in traffic signal control). The discount factor $\gamma \in [0, 1]$ balances immediate and future rewards. The expectation is estimated via multiple rollouts to account for stochasticity and ensure the reliability of the ground-truth action.

### 4.1.5 REFLECTION QA

Unlike static problem-solving (*e.g.,* mathematics), urban systems continuously generate time-indexed feedback (*e.g.,* traffic states at future timesteps) that reveals the consequences of prior decisions. Solving urban tasks requires not only executing accurate actions at each step, but also diagnosing and correcting past potential mistakes with feedback signals, then adapting strategies to the evolving urban environment (Ji et al., 2023; Kim et al., 2025).

To evaluate reflection reasoning, we construct QA instances in which the agent is provided with a prior action or prediction (*e.g.,* $a_{i-1}$ or $s_{i-1}$), the current observation $o_i$, and the resulting environmental feedback $f_i$. The agent is tasked with assessing its prior output and determining whether the previous decision or prediction was appropriate. If not, it needs to revise its earlier outputs accordingly. Performance is measured by the accuracy of corrections. This setting explicitly tests whether the agent can transform external feedback into causal links between its past actions and observed outcomes, then leverage this understanding to improve and adjust strategies in subsequent steps.

### 4.2 END-TO-END DOWNSTREAM TASK EVALUATION

To directly evaluate LLMs' performance in real-world applications, we conduct end-to-end assessments across nine urban tasks with UAgentEnv. Equipped with our urban agent framework, each task is uniformly evaluated using domain-specific metrics. For example, we use Mean Absolute Percentage Error (MAPE) to assess GDP forecasting accuracy over a three-year window in socioeconomic prediction. Congestion prediction, which classifies congestion into five levels (0 to 4), is evaluated using accuracy and MAPE. Urban planning performance is assessed based on two criteria: accessibility to service facilities and ecological coverage. For road planning, we measure construction costs and the average travel distance to neighboring regions. The tasks mentioned above are assessed in 5.3. Appendix D details the evaluation metrics and results of the remaining five tasks.

## 5 EXPERIMENT

### 5.1 BASELINE MODELS

We evaluate both non-reasoning and reasoning models with the same parameter sizes and architectures (*e.g.,* Qwen2.5-32B vs. QwQ-32B) to isolate the effectiveness of the improvement in reasoning abilities. For non-reasoning LLMs, we include: Qwen2.5 (7B and 32B) (Yang et al., 2024), GLM4 (9B and 32B) GLM et al. (2024), Llama3.3-70B Grattafiori et al. (2024), and GPT-4o (Hurst et al., 2024). For reasoning models, we evaluate: DeepSeek-R1-Distill (7B and 70B), DeepSeek-R1 (Guo

Table 3: Performances on spatiotemporal understanding.

| Model | Spatial Understanding | | | | Temporal Understanding | | | | | | Overall |
|---|---|---|---|---|---|---|---|---|---|---|---|
| | Distance | Adjacency | Connectivity | Overall | Duration | Local Extrema | Chronology | Periodicity | Trend | Overall | |
| Random | 0.2500 | 0.2500 | 0.2500 | 0.2500 | 0.2500 | 0.2500 | 0.2500 | 0.2500 | 0.1100 | 0.2200 | 0.2344 |
| *Non-Reasoning LLMs* | | | | | | | | | | | |
| Qwen2.5-7B | 0.5080 | 0.4070 | 0.2513 | 0.3888 | 0.5902 | 0.5767 | 0.4710 | 0.6637 | 0.1740 | 0.4951 | 0.4552 |
| GLM4-9B | 0.5389 | 0.4400 | 0.2993 | 0.4261 | 0.6522 | 0.6362 | 0.5354 | 0.6438 | 0.1611 | 0.5257 | 0.4883 |
| Qwen2.5-32B | 0.8046 | 0.5623 | 0.4537 | 0.6068 | 0.8136 | 0.7303 | 0.6610 | 0.7923 | 0.1613 | 0.6317 | 0.6224 |
| GLM4-32B | 0.7541 | 0.6618 | 0.4555 | 0.6238 | 0.7662 | 0.7513 | 0.6414 | 0.7248 | 0.1742 | 0.6116 | 0.6162 |
| Llama3.3-70B | 0.7448 | 0.5978 | 0.5113 | 0.6180 | 0.8148 | 0.7540 | 0.6630 | 0.7163 | 0.3203 | 0.6537 | 0.6403 |
| GPT-4o | 0.9295 | 0.6963 | 0.6787 | 0.7681 | 0.9288 | 0.8260 | 0.7310 | 0.8063 | 0.2110 | 0.7006 | 0.7259 |
| *Reasoning LLMs* | | | | | | | | | | | |
| DeepSeek-R1-7B | 0.4386 | 0.0450 | 0.0337 | 0.1724 | 0.5254 | 0.2820 | 0.389 | 0.3773 | 0.2580 | 0.3663 | 0.2936 |
| GLM-Z1-9B | 0.8023 | 0.6929 | 0.5627 | 0.6860 | 0.8126 | 0.8271 | 0.6717 | 0.8234 | 0.3855 | 0.7041 | 0.6973 |
| gpt-oss-20B | 0.9378 | 0.8444 | 0.4922 | 0.7581 | 0.9854 | 0.8563 | 0.7273 | 0.8357 | **0.4612** | **0.7732** | 0.7675 |
| QwQ-32B | 0.9508 | 0.7875 | 0.7450 | 0.8278 | 0.9818 | 0.8490 | 0.6810 | 0.8433 | 0.2777 | 0.7266 | 0.7645 |
| GLM-Z1-32B | 0.9053 | 0.8022 | 0.5978 | 0.7684 | 0.9053 | 0.8655 | 0.7071 | 0.8162 | 0.2555 | 0.7099 | 0.7319 |
| DeepSeek-R1-70B | 0.9528 | 0.6618 | 0.6867 | 0.7671 | 0.9500 | 0.8440 | 0.6850 | 0.7850 | 0.4280 | 0.7384 | 0.7492 |
| DeepSeek-R1 | 0.9310 | **0.8598** | 0.6808 | 0.8239 | 0.9492 | **0.8902** | 0.7374 | 0.8540 | 0.2502 | 0.7362 | 0.7691 |
| o4-mini | **0.9798** | 0.8597 | **0.7665** | **0.8687** | 0.9930 | 0.8884 | **0.7475** | 0.8704 | 0.2340 | 0.7467 | **0.7924** |

et al., 2025), QwQ-32B (Team, 2025), GLM-Z1 (9B and 32B) (GLM et al., 2024), gpt-oss-20B (Agarwal et al., 2025), and o4-mini (Jaech et al., 2024). DeepSeek-R1 provides multiple reasoning distillation variants with the same architectures and sizes, enabling fair comparisons across scales. GPT-4o and o4-mini state the state-of-the-art closed-source non-reasoning and reasoning LLMs. To contextualize LLM performance in end-to-end downstream tasks, we further include traditional domain-specific baselines. Details are provided in Appendix D.2.

## 5.2 SPATIOTEMPORAL REASONING QA EVALUATION

### 5.2.1 SPATIOTEMPORAL UNDERSTANDING

Table 3 reports model performance on spatiotemporal understanding. The evaluation results have been demonstrated to be consistent with the human evaluation. Details are in Appendix D.3. Overall, LLMs excel in interpreting urban spatiotemporal relations and patterns, with all models significantly surpassing the random baseline. Reasoning models achieve over 80% accuracy across multiple abilities. This indicates that LLMs' broad pretraining has embedded transferable priors for urban spatiotemporal reasoning. Additionally, their performance is notably stronger in spatial understanding, reflecting the greater complexity and variability of temporal dynamics compared to static spatial structures. Despite these advantages, LLMs struggle with long-range spatial relations (*i.e.,* connectivity) and long-term temporal patterns, including event chronology, and time-series periodicity or trends (*e.g.,* traffic flow shifts), which often fall below 70% accuracy. These limitations likely stem from the pretraining on large-scale unstructured text, which provides limited exposure to multi-step reasoning over relational or sequential data. Moreover, DeepSeek-R1-7B shows a marked drop due to repetition issues (Li et al., 2023). Further analysis and more cases are provided in the Appendix D.7.2-D.7.3.

**Non-Reasoning LLMs vs. Reasoning LLMs**: Reasoning models like DeepSeek-R1, QwQ, and GLM-Z1 generally outperform their non-reasoning versions with gains of 7–20%, highlighting the benefits of reasoning-focused post-training. However, their advantage is not consistent. Firstly, GPT-4o often matches or exceeds models like GLM-Z1-32B and DeepSeek-R1-Distill-70B, suggesting that general post-training on logical and mathematical problems does not always benefit urban spatiotemporal reasoning. Notably, as stated in Section 1, DeepSeek-R1 falls short in understanding temporal trends, exposing its limitations in urban dynamic analysis. In contrast, Llama3.3-70B excels in this ability among non-reasoning models. Interestingly, this strength has been successfully transferred to DeepSeek-R1-70B, which is post-trained on Llama3.3 and excels in this task. This motivates us to explore domain-adaptive methods to improve these abilities.

### 5.2.2 FORECASTING AND PLANNING

Table 4 summarizes model performance on forecasting and planning. The results show that most LLMs exhibit promising forecasting capabilities, with leading models achieving accuracy above 70%. However, in tasks involving long-term temporal trend analysis (*e.g.,* congestion and traffic-OD prediction), non-reasoning base models outperform reasoning variants (*e.g.,* Qwen-2.5 vs. QwQ, Llama3.3 vs. DeepSeek-R1). This further suggests that the general enhancement of reasoning abilities

Table 4: Performance of LLMs in forecasting, planning, and reflection abilities.

| Model | Forecasting | | | | | Planning | | | | | | Reflection |
|---|---|---|---|---|---|---|---|---|---|---|---|---|
| | Next POI Prediction | Socio-economic Prediction | Congestion Prediction | Traffic-OD Prediction | Overall | Traffic Signal Control | POI Placement | Road Planning | Route Planning | Urban Planning | Overall | |
| Random | 0.25 | 0.25 | 0.25 | 0.25 | 0.25 | 0.25 | 0.25 | 0.25 | 0.25 | 0.25 | 0.25 | 0.25 |
| **Non-Reasoning LLMs** | | | | | | | | | | | | |
| Qwen2.5-7B | 0.4640 | 0.4879 | 0.6463 | 0.3907 | 0.4972 | 0.3760 | 0.2190 | 0.2960 | 0.3490 | 0.2620 | 0.3004 | 0.1899 |
| GLM4-9B | 0.5660 | 0.6456 | 0.6880 | 0.4690 | 0.5922 | 0.3920 | 0.2890 | 0.2930 | 0.4430 | 0.2480 | 0.3330 | 0.2758 |
| Qwen2.5-32B | 0.7473 | 0.8449 | 0.6900 | 0.4930 | 0.6938 | 0.5227 | 0.2047 | 0.3030 | 0.5483 | 0.3430 | 0.3843 | 0.3184 |
| GLM4-32B | 0.7090 | 0.7812 | 0.6730 | 0.4640 | 0.6568 | 0.5550 | 0.2230 | 0.3910 | 0.4740 | 0.3010 | 0.3888 | 0.2087 |
| Llama3.3-70B | 0.7507 | 0.6301 | **0.7493** | 0.5110 | 0.6603 | 0.5313 | 0.3273 | 0.3563 | **0.6950** | 0.3413 | 0.4503 | 0.1935 |
| GPT-4o | 0.8280 | 0.9029 | 0.7380 | 0.4840 | 0.7382 | 0.5620 | 0.2820 | 0.3330 | 0.4780 | 0.3380 | 0.3986 | 0.3802 |
| **Reasoning LLMs** | | | | | | | | | | | | |
| DeepSeek-R1-7B | 0.4990 | 0.2488 | 0.4597 | 0.2193 | 0.3567 | 0.4120 | 0.1507 | 0.2000 | 0.4117 | 0.2493 | 0.2847 | 0.1068 |
| GLM-Z1-9B | 0.8740 | 0.9159 | 0.6020 | 0.4280 | 0.7050 | 0.5280 | **0.4990** | 0.3960 | 0.6000 | 0.3160 | **0.4678** | 0.4293 |
| gpt-oss-20B | 0.8780 | 0.9612 | 0.7110 | 0.4940 | 0.7611 | **0.5970** | 0.3560 | 0.3640 | 0.3390 | 0.3180 | 0.3948 | 0.4873 |
| QwQ-32B | 0.9153 | 0.9015 | 0.6290 | 0.4987 | 0.7361 | 0.5637 | 0.4897 | 0.3587 | 0.4223 | 0.3997 | 0.4468 | 0.4804 |
| GLM-Z1-32B | 0.9230 | 0.8957 | 0.6690 | 0.4360 | 0.7309 | 0.5520 | 0.4100 | 0.3910 | 0.3660 | 0.3120 | 0.4062 | 0.4597 |
| DeepSeek-R1-70B | 0.8737 | 0.8691 | 0.5947 | **0.5280** | 0.7164 | 0.5173 | 0.3227 | 0.3703 | 0.3517 | 0.3763 | 0.4035 | 0.4035 |
| DeepSeek-R1 | 0.8900 | 0.8706 | 0.6020 | 0.4780 | 0.7101 | 0.5160 | 0.3120 | 0.4060 | 0.3120 | 0.4380 | 0.3968 | **0.5179** |
| o4-mini | **0.9320** | **0.9709** | 0.7360 | 0.5100 | **0.7872** | 0.5860 | 0.3420 | **0.4610** | 0.3840 | **0.4600** | 0.4466 | 0.5011 |

Figure 4: The performance of the model with enhanced spatiotemporal understanding abilities.

does not always benefit urban scenarios. Meanwhile, the planning performance is substantially lower, revealing that current LLMs struggle to make accurate decisions aligned with long-term objectives. This disparity highlights the increased complexity of planning and reinforces our claim in Section 4.1.3 that planning is a higher-order ability, dependent on and extending beyond forecasting.

**Interplay Between Reasoning Abilities**: Models that excel in spatiotemporal understanding generally perform better in downstream reasoning. For example, gpt-oss-20B exhibits superior long-horizon temporal understanding abilities (*e.g.,* chronology, periodicity, and trend, as shown in Table 3), which directly enhances its performance in long-term temporal forecasting (*e.g.,* socio-economic, congestion, and traffic-OD prediction). To validate this connection, we post-train Qwen2.5-7B specifically on spatiotemporal understanding (the training details are in Appendix C). As illustrated in Figure 4, the fine-tuned model (*i.e.,* Qwen2.5-7B-ST) not only significantly outperforms its base (*i.e.,* Qwen2.5-7B) but reasoning (*i.e.,* DeepSeek-R1-Distill-Qwen-7B) variant, confirming the benefit of improved spatiotemporal understanding for forecasting and planning. Future research could focus on: (1) improving LLMs' comprehension of structured spatiotemporal data, and (2) integrating tools and code-based modeling to support more robust spatiotemporal pattern interpretation.

### 5.2.3 REFLECTION

Table 4 shows that most LLMs achieve less than 50% accuracy on reflection, indicating that existing models struggle to interpret feedback and diagnose potential errors. Reflection is essential for generalizing experience to subsequent reasoning, a high-level ability that facilitates long-term adaptation in dynamic urban environments. Its impact on downstream tasks is further analyzed in Section 5.3.

**Limitations in Reflection Ability**: We further conduct a qualitative study of reflection errors using GPT-5 (OpenAI, 2025) as an LLM-based judge. We examine three models with varying reflection capabilities (*i.e.,* Qwen2.5-7B, Qwen2.5-32B, and DeepSeek-R1) and categorize reflection performance across three aspects: (1) *Error types*, including feedback interpretation, adaptation, and integration errors; (2) *Reasoning quality*, ranging from complete failure to detect errors and fixing them; and (3) *Faithfulness*, identifying behaviors such as overconfident but wrong reflections or inconsistent reasoning. As shown in Figure 5, DeepSeek-R1 consistently demonstrates stronger reflection ability, exhibiting fewer interpretation and adaptation errors and achieving a higher correct-fix rate. However, it shows consistent challenges in dynamically integrating feedback, indicating that improved reasoning alone does not eliminate brittleness in adaptation. We also observe that non-reasoning models often exhibit overconfidence even when their conclusions are incorrect. In

Figure 5: The analysis of error type, reasoning quality, and faithfulness on reflection.

Table 5: Downstream tasks performance. Lower (↓) MAPE, Cost, and Distance (Dis.) are better, while higher (↑) Accuracy (Acc.), Service (Ser.), and Ecology (Eco.) are better.

| Model | Socio-economic Prediction | Congestion Prediction | | Urban Planning | | Road Planning | |
|---|---|---|---|---|---|---|---|
| | MAPE (↓) | MAPE (↓) | Acc. (↑) | Ser. (↑) | Eco. (↑) | Cost (↓) | Dis. (↓) |
| Classic Method | 7.09% | 57.05% | 17.18% | 0.6100 | 0.4310 | 18.95 | 1.99 |
| **Non-Reasoning LLMs** | | | | | | | |
| Qwen2.5-7B | 34.57% | 66.19% | 40.51% | 0.5951 | 0.6440 | 20.72 | 1.50 |
| GLM4-9B | 58.43% | 41.41% | 54.71% | 0.6355 | 0.4507 | 20.59 | 1.50 |
| Qwen2.5-32B | 6.00% | 24.90% | 65.90% | 0.6335 | 0.5209 | 20.56 | 1.55 |
| GLM4-32B | 9.41% | 28.61% | 63.02% | 0.6662 | 0.4715 | 18.44 | 1.52 |
| Llama3.3-70B | 10.86% | 38.88% | 56.10% | 0.6561 | 0.5842 | 19.10 | 1.57 |
| **Reasoning LLMs** | | | | | | | |
| DeepSeek-R1-7B | 79.23% | 67.42% | 37.88% | 0.6348 | 0.6111 | 20.60 | 1.47 |
| GLM-Z1-9B | 11.58% | 45.87% | 52.01% | 0.6443 | 0.5430 | 18.80 | 1.33 |
| gpt-oss-20B | 5.00% | 23.26% | 67.41% | 0.5842 | 0.5222 | 18.67 | 1.64 |
| QwQ-32B | 5.64% | 44.89% | 52.88% | 0.6751 | 0.5792 | **18.40** | 1.77 |
| GLM-Z1-32B | 7.55% | 47.93% | 51.22% | 0.6468 | 0.3965 | 18.57 | 1.87 |
| DeepSeek-R1-70B | 5.94% | 38.78% | 55.50% | 0.6560 | 0.4711 | 19.42 | **1.13** |
| DeepSeek-R1 | 5.24% | 41.38% | 58.75% | **0.6858** | **0.6651** | 18.49 | 1.86 |
| o4-mini | **4.97%** | **15.78%** | **75.73%** | 0.6544 | 0.3863 | 19.60 | 1.23 |

comparison, DeepSeek-R1 exhibits occasional inconsistencies, suggesting that faithfulness is an orthogonal challenge that is not resolved by enhanced reasoning abilities in general domains.

## 5.3 DOWNSTREAM TASK EVALUATION

Table 5 presents the performance of LLMs on four real-world urban tasks. Generally, LLMs outperform classic methods across prediction and decision-making tasks. Notably, we observe performance improvements up to 337.31% in forecasting accuracy and 53.48% in decision effectiveness. This reinforces the promise of LLMs as flexible and robust agents for various urban spatiotemporal tasks. Similar to previous observations, models like Qwen2.5-32B and LLaMA3.3 outperform their reasoning variants (*i.e.,* QwQ and DeepSeek-R1) on long-term prediction tasks (*i.e.,* congestion prediction). This trend highlights the potential bottlenecks of reasoning LLMs in urban spatiotemporal reasoning.

**The Impact of Different Reasoning Abilities**: To evaluate the contribution of individual reasoning components to overall agent performance, we perform ablations in descending order of model reasoning strength, as shown in Figure 6. For DeepSeek-R1, the top-tier reasoning model, removing spatiotemporal understanding significantly increases prediction error and reduces downstream planning accuracy, indicating that the model relies heavily on accurate early interpretation to support subsequent reasoning. Eliminating forecasting similarly degrades planning, demonstrating that DeepSeek-R1 effectively uses predictions to inform long-term decision-making. Reflection removal produces the largest performance drop, confirming that the model can meaningfully leverage feedback to refine its outputs. In contrast, models with medium reasoning abilities (*e.g.,* Qwen2.5-32B) exhibit moderate sensitivity. While removing spatiotemporal understanding still harms performance, bypassing forecasting leads to a slight improvement in planning, suggesting that noisy intermediate predictions may misguide downstream reasoning. For weaker models (*e.g.,* Qwen2.5-7B), intermediate reasoning and reflection can even be detrimental, as limited reasoning capacity produces unreliable outputs that propagate additional error rather than providing corrective guidance.

## 6 RELATED WORK

**Urban Agent**: Urban agents have evolved from rule-based systems to RL agents and now LLM-based models. Early systems like SCOOT (Hunt et al., 1982) and SCATS (Lowrie, 1990) used fixed heuristics and sensor data for traffic control. RL approaches, such as CoLight (Wei et al., 2019),

Figure 6: The ablation of each reasoning ability on the task performance.

introduced data-driven adaptive learning using neural networks. Recently, LLM-based agents, like LLMLight (Lai et al., 2025), UrbanGPT (Li et al., 2024c), and UrbanKGent (Ning & Liu, 2024), leverage LLMs for tasks like traffic optimization, spatiotemporal forecasting, and urban knowledge base construction, enabling more flexible and scalable urban intelligence. LLMob (Wang et al., 2024a) further frame LLMs as "urban residents" to generate realistic mobility trajectories and activity schedules, advancing agent-level behavior modeling. In contrast, UrbanLLM (Jiang et al., 2024) fine-tunes LLMs to decompose urban queries and orchestrate end-to-end planning autonomously.

**Spatiotemporal Reasoning**: Spatial reasoning in LLMs has been assessed by understanding spatial relations (Mirzaee et al., 2021). Later works introduced multi-hop reasoning tasks (Shi et al., 2022) and urban challenges (Zhan et al., 2025). Notably, CityEQA (Zhao et al., 2025) extends spatial reasoning to embodied agents navigating city spaces. Temporal reasoning focuses on understanding event sequences. Studies like (Fatemi et al., 2024; Chu et al., 2023) examined commonsense temporal reasoning. Spatiotemporal reasoning integrates both spatial and temporal dynamics. Recent work (MOONEY et al., 2023; Quan et al., 2025) analyzed how LLMs interpret spatiotemporal patterns. In urban contexts, STBench (Li et al., 2024b) and CityBench (Feng et al., 2025b) evaluate the reasoning abilities on trajectories and interactions between spatial entities over time. Moreover, LogiCity (Li et al., 2024a) provides a neuro-symbolic urban simulator for compositional, long-horizon reasoning.

**LLM Complex Reasoning**: LLM reasoning has seen rapid advances through chain-of-thought (Wei et al., 2022), which improves multi-step problem solving by encouraging intermediate reasoning. First, instruction tuning approaches, such as AgentTuning (Zeng et al., 2023), boost reasoning quality. Recently, post-training strategies adopted by models like OpenAI-o1 (Jaech et al., 2024), DeepSeek-R1 (Guo et al., 2025), and QwQ (Team, 2025) incorporate RL like Proximal Policy Optimization (PPO) (Schulman et al., 2017) and Group Relative Policy Optimization (GRPO) to refine reasoning through reward-guided optimization. Recent advances extend LLMs into agents operating in complex environments. PERIA (Ni et al., 2024) enhances context understanding and forecasts intermediate states to drive long-horizon planning. PreAct (Fu et al., 2025) strengthens planning by integrating explicit future-state prediction. ReflAct (Kim et al., 2025) improves reliability through goal-state reflection during action execution. MoSciBench (Liu et al., 2026) introduces the first multimodal scientific discovery benchmark, exposing cross-modal reasoning challenges. MM-Agent (Liu et al., 2025b) states a four-stage LLM agent framework for open-ended real-world mathematical modeling.

# 7 CONCLUSION AND LIMITATIONS

**Conclusion**: We present USTBench, the first benchmark for systematically evaluating the spatiotemporal reasoning abilities of LLMs as urban agents. Built on the interactive environment UAgentEnv, USTBench supports both fine-grained diagnostics of specific reasoning processes and standardized end-to-end task performance evaluations. Our evaluation of leading LLMs shows that while current models excel in spatiotemporal understanding and forecasting, they struggle with higher-order reasoning abilities, particularly long-term planning and reflection. Notably, reasoning models trained on general logic and mathematics do not consistently outperform non-reasoning models in urban-specific tasks, underscoring the importance of domain adaptation for further enhancement.

**Limitations**: USTBench primarily focuses on evaluation, while methods for enhancing spatiotemporal reasoning are underexplored. Moreover, while our study establishes a core, interpretable, and empirically testable decomposition to measure LLM agent performance in spatiotemporal reasoning, it does not attempt to cover all aspects of LLM agents relevant to broader urban tasks (*e.g.,* social reasoning, multi-agent interaction). Additionally, the evaluations on decision-making tasks are mainly based on simulated environments, while real-world validation is essential for urban applications. Future work will explore real-world experiments on decision-making to address this gap.

## ACKNOWLEDGMENTS

This work was supported by the National Natural Science Foundation of China (Grant No. 62572417, No.92370204), National Key R&D Program of China (Grant No.2023YFF0725004), and CCF-DiDi GAIA Collaborative Research Funds.

## ETHICS STATEMENT

USTBench aims to systematically evaluate the spatiotemporal reasoning capabilities of LLMs in urban applications such as traffic control, mobility prediction, and urban planning. To ensure privacy, our benchmark exclusively uses publicly available datasets and ensures that no personally identifiable information is included. While these evaluations advance the understanding of LLM capabilities, they also underscore the responsibility to ensure that such models are used ethically. LLM-driven urban agents could influence public infrastructure, mobility patterns, and access to services. Therefore, deploying these models without proper oversight or fail-safes could lead to unintended negative outcomes, especially for vulnerable populations in urban settings.

## REPRODUCIBILITY STATEMENT

We have open-sourced all environment configurations, benchmark datasets, and evaluation scripts at `https://github.com/usail-hkust/USTBench`. The core methodology for spatiotemporal reasoning evaluation using UAgentEnv is described in Sections 3–5. To ensure reproducibility, we provide comprehensive experimental details, including LLM configurations, baseline implementations (Appendix D.2), and evaluation metrics (Appendix D.1). The 62,466 structured QA pairs for process-based evaluation were generated through systematic interactions with UAgentEnv, with representative samples shown in Appendix D.7.3. Data processing pipelines are documented in Appendices C.2, B.1, and B.2. Human evaluation procedures and results are reported in Appendix D.3. Finally, the experimental platform and runtime statistics are detailed in Appendix D.5.

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

# USTBench: Benchmarking and Dissecting Spatiotemporal Reasoning Capabilities of LLMs as Urban Agents

## *Supplementary Material*

CONTENTS

## A  LLM USAGE

We employ LLMs to enhance the clarity and coherence of our manuscript writing, ensuring consistency in terminology, structure, and academic tone. In addition, LLMs are utilized to generate figures and illustrations that effectively visualize experimental results, improving the overall readability and presentation quality of the paper.

## B  DETAILS OF ENVIRONMENTS

### B.1  URBAN TASK SUITE AND ENVIRONMENT PLATFORM

**Next POI Prediction**: Predicts movement patterns within urban areas, helping to understand how individuals travel between points of interest (POIs). Following the settings proposed by (Wang et al., 2023), we use FourSquare datasets (Yang, 2014) for evaluation.

**Socio-Economic Prediction**: Estimates future socio-economic indicators (*i.e.,* GDP) based on historical observations and population density. We collect global GDP and population data from 2000 to 2019 (Wang & Sun, 2022; WorldPop, 2025) for future value prediction.

**Congestion Prediction**: Anticipates areas where traffic flow will become heavy. We follow the settings of (Cheng et al., 2018) and evaluate LLMs using traffic data from Beijing, China.

**Traffic-OD Prediction**: Estimates vehicle flow between origin and destination regions. We follow the settings of (Yuan et al., 2024) and use taxi traffic flow datasets from New York, USA.

**Traffic Signal Control**: Optimizes traffic signal timing to improve traffic flow. Following the settings of (Lai et al., 2025), we use traffic flow and road network datasets from China and the US (Zhang et al., 2023). The simulation environment is built based on CityFlow (Tang et al., 2019).

**POI Placement**: Determines optimal locations for urban services (*e.g.,* shops, restaurants), ensuring strategic placement to serve the population and reduce congestion. For evaluation, we follow the settings of (von Wahl et al., 2022) and collect the POI dataset in Guangzhou, China.

**Road Planning**: Involves designing and optimizing road networks to enhance transportation efficiency. This task requires analyzing road connectivity and infrastructure needs. We follow the settings and environment of (Zheng et al., 2023b) and use road network data from Cape Town (South Africa), Harare (Zimbabwe), and Mumbai (India) for evaluation.

**Route Planning**: Determines optimal paths for vehicles, considering traffic conditions, distance, and travel time. Following the settings of (Li et al., 2021), we use New York's road network and simulate traffic using the SUMO traffic simulator (Lopez et al., 2025).

**Urban Planning**: Involves designing urban spaces for sustainable and efficient cities. Following (Zheng et al., 2023a), we use urban geospatial data from Beijing, China.

### B.2  ENVIRONMENT CONFIGURATION

For process-based spatiotemporal reasoning evaluation, we generate QA instances using UAgentEnv. Environmental observations in decision-making tasks are collected using a semi-stochastic policy with an exploration coefficient $\epsilon = 0.1$ to ensure diversity. Ground-truth answers for planning QA are derived through a simulation-driven exploratory process, with a planning horizon $H = 5$ and a discount factor $\gamma = 0.9$. During downstream task evaluation, we configure historical observation and prediction windows based on task characteristics. For socio-economic prediction, we use a 6-year observation and a 3-year prediction window. For traffic-related tasks—congestion prediction and traffic OD prediction, we adopt a 12-step observation and a 12-step prediction window. In the next POI prediction task, the agent receives a 30-day activity history and recent visits on the same day and predicts the next visited POI. For route planning, we synthesize urban mobility patterns using the gravity model (Li et al., 2021), calibrated on real-world population distributions. The configurations for decision-making environments follow established benchmarks and experimental protocols from prior work (Wei et al., 2019; von Wahl et al., 2022; Li et al., 2021; Zheng et al., 2023b;a).

### B.3  AGENT PROMPT

Our agent framework incorporates three prompt templates designed for task-solving, reflection, and memory storage. In the *task-solving prompt*, the agent is instructed to perform a multi-stage

reasoning process: 1) it first interprets the spatiotemporal context of the environment; 2) then either forecasts future urban states or predicts the outcomes of candidate actions; 3) finally, the agent outputs anticipated urban states for prediction tasks or selects an optimal action for decision-making tasks. After execution and receiving environmental feedback, the agent is instructed by the *reflection prompt* to evaluate the effectiveness of its prior decision and summarizes the outcome as an experience to inform future reasoning. These experiences are subsequently aggregated and stored in memory to support continual adaptation with the *summary prompt*. The detailed prompt templates used in each stage are provided in Table 21-23.

## C   LLM Post-Training

### C.1   Instruction Construction

We post-train Qwen2.5-7B using a synthetic instruction tuning dataset, designed to enhance the model's capability to interpret spatiotemporal dynamics in urban scenarios. To generate instructions, we first prompt GPT-4o to produce diverse urban scenarios involving various entities and events. Each scenario is designed to elicit a specific spatiotemporal understanding ability, prompting the model to analyze spatial relationships or temporal patterns:

**Distance**: We randomly assign distances between roads, routes, or urban entities. The model is asked to identify the longest or shortest elements or to compare distances between pairs of entities.

**Adjacency & Connectivity**: We Erdős–R'enyi (ER) model (Erdos et al., 1960) to generate random spatial graphs representing urban layouts. The model is then asked to determine adjacency (*i.e.,* nearby neighbors) or connectivity (*i.e.,* path existence between entities).

**Duration**: We simulate urban events (*e.g.,* travel, wait times) with randomly assigned durations. The model is tasked with identifying the longest or shortest event or comparing durations between events.

**Chronology**: We use POI check-in data from Tokyo (Yang, 2014) (distinct from the dataset used in USTBench). The model is tasked to identify the correct temporal sequence of check-in events.

**Trend, Local Extrema, and Periodicity**: We leverage real-world urban time-series datasets, including PEMS04 (traffic flow), Solar (solar power output), and Electricity (power usage). The model is instructed to identify global trends (*e.g.,* increasing/decreasing), local extrema (*e.g.,* peak hours), and periodicity (*e.g.,* daily/weekly cycles).

### C.2   Supervised Distillation Fine-Tuning

Leveraging the constructed instructions described above, we collect responses from DeepSeek-R1 using rejection sampling to ensure high-quality outputs. Representative examples of the instruction tuning data are provided in Table 24-31. For supervised fine-tuning, we adopt Llama-Factory (Zheng et al., 2024) with Low-Rank Adaptation (LoRA) for training. The learning rate is set to $1 \times 10^{-4}$.

## D   Detailed Evaluations

### D.1   Downstream Task Evaluation Metrics

We evaluate LLMs' downstream task performance across nine urban tasks using task-specific metrics. *Socio-economic prediction*: We apply Mean Absolute Percentage Error (MAPE) to assess GDP forecasting accuracy over a three-year window. *Congestion prediction*: This task is framed as a five-level classification task (levels 0–4), and is evaluated using MAPE and accuracy. *Traffic-OD prediction*: We adopt Mean Absolute Error (MAE) and Symmetric Mean Absolute Percentage Error (SMAPE) to evaluate forecasting accuracy for vehicle arrivals and departures. *Next POI prediction*: We use Precision and Mean Reciprocal Rank (MRR) to measure the recall quality of predictions. *Urban planning*: This task is assessed through accessibility to service facilities and ecological coverage. *Road planning*: The performance is measured by construction costs and the average travel distance to neighboring regions. *POI placement*: The task is evaluated by the average travel and waiting time for urban services. *Traffic signal control*: The performance is measured using average travel time (ATT) and waiting time (AWT) in the road network. *Route planning*: The task is evaluated using average travel time and network throughput.

Table 6: Comparison of exact-match and human evaluation across spatiotemporal reasoning tasks.

| Model | Evaluation Method | Spatiotemporal Understanding | Forecasting | Planning | Reflection |
|---|---|---|---|---|---|
| Llama3.3-70B | Exact-match | 0.6000 | 0.6500 | 0.4800 | 0.2222 |
| | Human-Eval | 0.6428 | 0.7000 | 0.5000 | 0.2556 |
| DeepSeek-R1 | Exact-match | 0.7000 | 0.6250 | 0.4400 | 0.4556 |
| | Human-Eval | 0.7572 | 0.7000 | 0.4800 | 0.5111 |

Table 7: Confidence Interval of Spatiotemporal Reasoning Evaluation with Accuracy (%).

| Model | Spatial Understanding | | | Temporal Understanding | | | | | Forecasting | Planning | Reflection |
|---|---|---|---|---|---|---|---|---|---|---|---|
| | Distance | Adjacency | Connectivity | Duration | Chronology | Trend | Local Extrema | Periodicity | | | |
| **Non-Reasoning LLMs** | | | | | | | | | | | |
| Qwen2.5-7B | 50.80 (±1.80) | 40.70 (±0.90) | 25.13 (±0.08) | 59.02 (±0.21) | 47.10 (±0.57) | 17.40 (±0.12) | 57.67 (±0.67) | 66.37 (±0.10) | 49.72 (±0.10) | 30.04 (±0.58) | 18.99 (±0.17) |
| Qwen2.5-32B | 80.46 (±0.28) | 56.23 (±0.73) | 45.37 (±4.10) | 81.36 (±0.13) | 66.10 (±1.23) | 16.13 (±0.03) | 73.03 (±0.29) | 79.23 (±0.66) | 69.38 (±0.23) | 38.43 (±5.53) | 31.84 (±0.10) |
| **Reasoning LLMs** | | | | | | | | | | | |
| DeepSeek-R1-Distill-Qwen-7B | 43.86 (±0.38) | 4.50 (±0.10) | 3.37 (±0.07) | 52.54 (±0.38) | 38.90 (±1.27) | 25.80 (±0.32) | 28.20 (±1.41) | 37.73 (±1.43) | 35.67 (±0.46) | 28.47 (±0.28) | 10.68 (±0.42) |
| QwQ-32B | 95.08 (±3.02) | 78.75 (±0.68) | 74.50 (±1.01) | 98.18 (±1.89) | 68.10 (±2.57) | 27.77 (±0.11) | 84.90 (±1.11) | 84.33 (±0.86) | 73.61 (±0.41) | 44.68 (±1.44) | 48.04 (±0.17) |

## D.2 BASELINE CONFIGURATION

**LLM Configuration**: In this study, we mainly evaluate open-source LLMs using the vLLM inference framework. For proprietary models such as GPT-4o and GPT-4o-mini, we utilize the OpenAI API, while DeepSeek-R1 is evaluated via the Alibaba Bailian API. All evaluations are conducted with a fixed decoding temperature of 0.1 to ensure reproducibility. Inference of open-sourced LLMs is performed on a server equipped with two NVIDIA A800-80GB GPUs.

**Domain-specific Method Configuration**: To provide a comprehensive performance comparison, we also benchmark LLMs against traditional methods widely used in each domain: For time-series forecasting tasks, we use ARIMA (Box) as the baseline model. For urban planning, we employ a geometric set coverage algorithm (GSCA) (Wei, 2016), which solves a geometric set-coverage-like problem by maximizing the spatial coverage of designated land-use types. For road planning, we apply a genetic algorithm (Gad, 2024), where a linear layer represents road features, and roads are incrementally constructed based on learned sampling probabilities.

## D.3 HUMAN EVALUATION

In our experiment, the LLM performance is primarily evaluated using exact-match comparison with the extracted ground-truth labels to ensure reproducibility and computational efficiency across large-scale evaluation. For completeness, we incorporate a human evaluation with five human experts in urban science over a randomly selected 250 QAs in our benchmark. This assessment examines not only the correctness of model answers but also evaluates the underlying reasoning process. The results are shown in Figure 6. We find that human evaluations generally align with our exact-match assessments, supporting the reliability of our automated metric.

## D.4 CONFIDENCE INTERVAL OF EVALUATION

In Table 7, we report the confidence intervals for representative models based on three experiments.

## D.5 RUNTIME ESTIMATION

The evaluation runtime of an LLM varies depending on the hardware or API, the specific model, and the inference platform employed. In this study, we estimate runtimes for open-source LLMs using vLLM, and for GPT-4o and GPT-4o-mini through the OpenAI API. Except for DeepSeek-R1, we evaluate with the Alibaba Bailian API. The estimated time of our process-based spatiotemporal reasoning evaluation is shown in Table 8.

## D.6 DOWNSTREAM TASK PERFORMANCE

The downstream task performance of representative LLMs is shown in Table 9. Notably, reasoning LLMs do not consistently outperform their non-reasoning base models in real-world urban scenarios. This indicates that advances in general mathematical and logical reasoning do not necessarily benefit urban tasks. The finding underscores the importance of developing domain-specific approaches tailored to the unique challenges of urban spatiotemporal reasoning.

Table 8: The runtime estimation results of the representative LLMs.

| Model | Device or API | Platform | Inference Speed | Batch Size | Total Time |
|---|---|---|---|---|---|
| Qwen2.5-32B | 2*A800 | vLLm | 14.12 s/batch | 32 | 9.9 h |
| GPT-4o | OpenAI API | - | 8.20 s/batch | 16 | 11.50 h |
| QwQ-32B | 2*A800 | vLLm | 30.21 s/batch | 32 | 18.64 h |
| DeepSeek-R1 | Alibaba Bailian API | - | 97.33 s/batch | 32 | 60.05 h |
| o4-mini | OpenAI API | - | 5.61 s/batch | 16 | 7.87 h |

Table 9: Performance on downstream urban tasks. Lower values (↓) for MAPE, SMAPE, MAE, Cost, Distance, ATT, and AWT indicate better performance. Higher values (↑) for Accuracy, Service, Ecology, Precision, MRR, and Throughput indicate better outcomes.

| Model | Socio-economy Prediction MAPE (%) | Congestion Prediction MAPE (%) | Congestion Prediction Acc (%) | Urban Planning Serv | Urban Planning Eco | Road Planning Cost | Road Planning Dist | Traffic-OD Prediction MAE | Traffic-OD Prediction SMAPE (%) | Next POI Prediction Prec@10 | Next POI Prediction MRR@10 | POI Placement ATT | POI Placement AWT | Signal Control ATT | Signal Control AWT | Route Planning ATT | Route Planning Thruput |
|---|---|---|---|---|---|---|---|---|---|---|---|---|---|---|---|---|---|
| **Non-Reasoning LLMs** | | | | | | | | | | | | | | | | | |
| Qwen2.5-7B | 34.57 | 66.19 | 40.51 | 0.5951 | **0.6440** | 20.72 | 1.50 | **5.09** | **13.35** | 0.3787 | 0.1888 | 1.21 | 0.56 | **820.28** | **472.20** | 1417.93 | 367 |
| GLM4-9B | 58.43 | 41.41 | 54.71 | 0.6355 | 0.4507 | 20.59 | 1.50 | 84.22 | 96.85 | **0.7017** | 0.4405 | 1.17 | 0.25 | 1109.80 | 626.89 | 1366.45 | 372 |
| Qwen2.5-32B | 6.00 | **24.90** | **65.90** | 0.6335 | 0.5209 | 20.56 | 1.55 | 8.11 | 33.43 | 0.6627 | 0.5096 | 1.16 | 0.47 | 1189.31 | 672.84 | 1376.33 | 376 |
| GLM4-32B | 9.41 | 28.61 | 63.02 | 0.6662 | 0.4715 | 18.44 | 1.52 | 11.20 | 43.70 | 0.5183 | 0.3786 | 1.12 | 0.36 | 1290.61 | 690.64 | 1384.54 | **373** |
| Llama3.3-70B | 10.86 | 38.88 | 56.10 | 0.6561 | 0.5842 | 19.10 | 1.57 | 8.52 | 33.13 | 0.4863 | 0.3273 | 1.12 | 0.36 | 1324.84 | 682.10 | 1310.11 | 370 |
| **Reasoning LLMs** | | | | | | | | | | | | | | | | | |
| DeepSeek-R1-Distill-Qwen-7B | 79.23 | 67.42 | 37.88 | 0.6348 | 0.6111 | 20.60 | 1.47 | 140.05 | 130.84 | 0.2910 | 0.1455 | 1.18 | 0.46 | 1000.56 | 541.84 | 1390.65 | 372 |
| GLM-Z1-9B | 11.58 | 45.87 | 52.01 | 0.6443 | 0.5430 | 18.80 | 1.33 | 70.03 | 92.38 | 0.5637 | 0.4963 | 1.17 | 0.52 | 970.26 | 711.33 | 1283.29 | 371 |
| QwQ-32B | **5.64** | 44.89 | 52.88 | **0.6751** | 0.5792 | **18.40** | 1.77 | 8.16 | 36.96 | 0.6817 | **0.6013** | 1.19 | 0.54 | 1267.82 | 672.88 | 1417.32 | 373 |
| GLM-Z1-32B | 7.55 | 47.93 | 51.22 | 0.6468 | 0.3965 | 18.57 | 1.87 | 56.46 | 77.46 | 0.6430 | 0.5540 | **1.10** | 0.74 | 1132.48 | 641.73 | 1331.51 | 370 |
| DeepSeek-R1-Distill-Llama-70B | 5.94 | 38.78 | 55.50 | 0.6560 | 0.4711 | 19.42 | **1.13** | 17.79 | 45.23 | 0.5530 | 0.4732 | 1.12 | **0.24** | 1202.11 | 629.29 | **1267.20** | 375 |

## D.7 REASONING BEHAVIOR ANALYSIS

### D.7.1 COST-EFFECTIVENESS ANALYSIS

In our process-based spatiotemporal reasoning evaluation, we observe that the non-reasoning model GPT-4o is comparable, and in some cases surpasses, reasoning models. To further explore this, we conduct a cost-effectiveness analysis (Figure 7), comparing model performance on spatiotemporal understanding relative to the number of reasoning tokens used. Among all models, o4-mini demonstrates the highest cost-efficiency, achieving strong performance with minimal reasoning overhead, followed closely by GPT-4o. In contrast, while DeepSeek-R1 delivers strong performance, its reasoning pro-

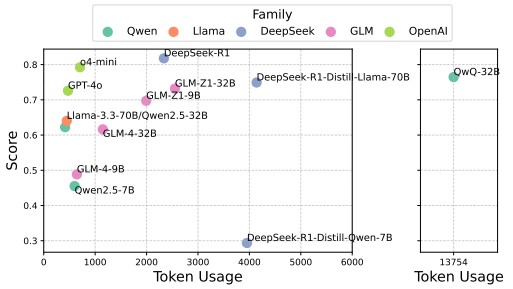

Figure 7: The token usage vs. score.

cesses are often verbose and time-consuming, making it less suitable for real-time deployment scenarios (*e.g.,* traffic management). These findings highlight research opportunities to develop lightweight and efficient paradigms for urban spatiotemporal reasoning, drawing inspiration from designs like o4-mini and GPT-4o.

### D.7.2 REPETITION ISSUES

We observe severe repetition issues on DeepSeek-R1-Distill-Qwen2.5-7B, where it underperforms its base model, Qwen2.5-7B, on several tasks. Upon analyzing its reasoning traces, we find that the model tends to repeat certain thought patterns, called "aha moments" (Guo et al., 2025), without progressing toward a solution. This repetitive behavior resembles ineffective in-context learning loops, and appears more pronounced in specialized domains like urban tasks. While such issues have been partially mitigated in mathematical and logical reasoning, they remain a persistent challenge in other specialized domains. Figure 19 provides examples of failed reasoning processes.

### D.7.3 PROCESS-BASED SPATIOTEMPORAL REASONING QA EXAMPLES

Table 10: The spatiotemporal understanding QA for distance relation analysis.

**Task Description**

We are tasked with determining the optimal location for a new electric vehicle (EV) charging station to maximize benefits by effectively meeting demand. The candidate locations, along with their expected charging station coverage, demand (number of vehicles), average distance, travel time, and waiting time to existing charging stations, are provided below.

**Data Schema**

- coverage: The number of locations that will be within the influence radius of the charging station after its construction.
- demand: The number of vehicles requiring charging near the location.
- distance: The average distance that vehicles near the location must travel to reach existing charging stations.
- waiting_time: The average waiting time at nearby existing charging stations. NaN indicates no station nearby and the highest waiting time.
- charging_time: The average charging time at nearby existing charging stations. NaN indicates no station nearby and the highest charging time.

**Data**

location 1:
- coverage: 3.315
- demand: 43531.73
- distance: 265.80m
- waiting_time: 0.00h
- charging_time: 1.80h

location 2:
- coverage: 3.316
- demand: 74504.47
- distance: 159.25m
- waiting_time: 0.04h
- charging_time: 20.61h

location 3:
- coverage: 3.316
- demand: 130231.14
- distance: 160.19m
- waiting_time: 0.04h
- charging_time: 21.75h

location 4:
...

**Question**

Based on the data provided, rank the locations by average distance to nearby charging stations, from the farthest to the closest.
A. 1 > 2 > 3 > 4
B. 2 > 4 > 3 > 1
C. 1 > 2 > 3 > 4
D. 4 > 1 > 3 > 2

**Note**

Please answer the question using the information given. Let's work this out in a step-by-step way to be sure that we have the right answer. Then provide your final answer within the XML tag
`<Answer>A/B/C/D</Answer>`.

✓ **Qwen2.5-32B**: The locations are ranked by average distance to nearby charging stations from the farthest to the closest as 4 > 1 > 3 > 2.

✓ **DeepSeek-R1**: The average distances for each location are: Location 4 (325.11m), Location 1 (265.80m), Location 3 (160.19m), and Location 2 (159.25m). Ranking from farthest to closest gives the order 4 > 1 > 3 > 2, which corresponds to option D.

✓ **o4-mini**: When ranking by average distance to existing charging stations from farthest to closest, the order is Location 4 (325.11 m) → Location 1 (265.80 m) → Location 3 (160.19 m) → Location 2 (159.25 m), which matches option D.

Figure 8: The LLMs' reasoning on distance relation analysis.

Table 11: The spatiotemporal understanding QA for adjacency relation analysis.

**Task Description**

You are tasked with predicting urban indicators—specifically, the GDP and population of Guangzhou for the current year. Historical data for the GDP and population of the target region and its nearby regions will be provided. Additionally, an example region with its GDP and population records for previous years and the current year will be included for reference.

**Data Schema**

- gdp_in_previous_years: The GDP (in millions of USD) of a region in past years.
- population_in_previous_years: The population of a region in past years.
- connectivity: A list of tuples in the format (region_1, region_2, distance), defining undirected edges between regions with the specified distance in m.

**Data**

*Target Region 442:*
- gdp_in_previous_years: [(8.54, 2000), ..., (23.19, 2018)]
- population_in_previous_years: [(2942.97, 2000), ..., (2242.3, 2018)]

*Nearby Region 418:*
- gdp_in_previous_years: [(22.55, 2000), ..., (55.37, 2018)]
- population_in_previous_years: [(2589.67, 2000), ..., (8729.58, 2018)]

...
*Adjacency Info:*
```
[(442, 418, 7.58km), (442, 443, 6.09km), (442, 467, 4.51km),
..., (467, 492, 4.51km)]
```
*Example Region 192:*
- gdp_records: [(201.59, 2000), ..., (628.55, 2019)]
- population_records: [(5908.03, 2000), ..., (11102.63, 2019)]

**Question**

Based on the connectivity data, rank spatial correlation from the target region to its nearby regions, ordering them from the strongest to the weakest correlation.
A. 492 > 468 > 417 > 443.
B. 443 > 492 > 468 > 417.
C. 492 > 468 > 417 > 443.
D. 468 > 443 > 492 > 417.

**Note**

Please answer the question using the information given. Let's work this out in a step-by-step way to be sure that we have the right answer. Then provide your final answer within the XML tag `<Answer>A/B/C/D</Answer>`.

✓ **Qwen2.5-32B**: The regions are ranked based on their shortest distance from the target region 442, with the closest regions having the strongest spatial correlation. The ranking from strongest to weakest correlation is: 467, 443, 418, 492, 468, 417. Option B.

✓ **DeepSeek-R1**: The spatial correlation is determined by the shortest distance from the target region (442) to each nearby region. Based on the adjacency data, the distances are calculated as follows: Region 443 (6.09km), Region 492 (9.02km via 467), Region 468 (10.59km via 443), and Region 417 (13.67km via 418 or 443). Ranking from strongest to weakest correlation (shortest to longest distance) gives the order: 443 > 492 > 468 > 417, which corresponds to option B.

✓ **o4-mini**: By computing shortest-path distances in the adjacency graph—442→443 (6.09 km), 442→467→492 (9.02 km), 442→443→468 (10.59 km), and 442→443→417 (13.67 km)—we rank regions by increasing distance (strongest to weakest spatial correlation) as 443 > 492 > 468 > 417. Option B.

Figure 9: The LLMs' reasoning on adjacency relation analysis.

Table 12: The spatiotemporal understanding QA for connectivity relation analysis.

**Task Description**

You are a traffic signal control agent at a target intersection, managing traffic from four directions: north, south, east, and west. The intersection has 12 lanes, categorized as: NL, NT, NR, SL, ST, SR, EL, ET, ER, WL, WT, and WR. Vehicles at the intersection travel at an average speed of 11 meters per second. The real-time traffic conditions for each lane are provided below. If a lane's traffic condition is not specified, it indicates that no vehicles are currently present in that lane.

**Data Schema**

- queue: Number of vehicles waiting at the lane for a green signal.
- move: Number of vehicles currently moving through the lane.
- wait_time: Average wait time (in minutes) for vehicles in the queue.
- occupancy: Lane occupancy rate (0 = free-flow, 100% = fully congested).
- connectivity: A list of tuples in the format (lane_1, lane_2, distance), defining a directed connection between lanes with the specified distance in m. The traffic flow from lane_1 (upstream) will arrive at lane_2 (downstream) in the future.

**Data**

Target Intersection 112:
- SL lane:
    - queue: 37.0
    - move: 0
    - wait_time: 2.86
    - occupancy: 112.12%
- ET lane:
    - queue: 18.0
    - move: 0
    - wait_time: 4.26
    - occupancy: 54.55%
...

Neighboring Intersection 140:
- ST lane:
    - queue: 4.0
    - move: 0
    - wait_time: 0.45
    - occupancy: 12.12%
- SL lane:
    - queue: 3.0
    - move: 1
    - wait_time: 0.41
    - occupancy: 12.12%
...
Other Neighbors...

The connectivity of lanes:
```
[(NT of 113, NT of 112, 300m), (EL of 113, NT of 112, 300m), (WR
of 113, NT of 112, 300m), (NT of 113, NL of 112, 300m), ..., (SR
of 84, WR of 112, 300m)]
```

**Question**

Based on the data provided, select the correct ranking of neighboring upstream lanes of the target intersection by the moving vehicles, ordering them from the highest to the lowest.
A. ST of 140 > EL of 140 = EL of 112 > other lanes
B. WR of 113 > ET of 84 = ET of 112 > other lanes
C. ET of 140 = ST of 140 > ET of 84 > other lanes
D. WR of 113 > ET of 140 = SL of 140 > other lanes

**Note**

Please answer the question using the information given. Let's work this out in a step-by-step way to be sure that we have the right answer. Then provide your final answer within the XML tag `<Answer>A/B/C/D</Answer>`.

> ✓ **Qwen2.5-32B**: The correct ranking of neighboring upstream lanes of the target intersection by the moving vehicles is WR of 113 > ET of 140 = SL of 140 > other lanes. Option D.
>
> ✓ **DeepSeek-R1**: The neighboring upstream lanes with moving vehicles are analyzed based on the connectivity and move counts. WR of 113 has the highest move count (5), followed by ET of 140 and SL of 140 (both 1). Other lanes have lower or zero move counts. Therefore, the correct ranking is WR of 113 > ET of 140 = SL of 140 > others, which matches option D. Option D.
>
> ✓ **o4-mini**: WR of 113 has the highest flow with 5 moving vehicles, ET of 140 and SL of 140 each have 1 moving vehicle (tied), and all other upstream lanes have none. Option D.

Figure 10: The LLMs' reasoning on connectivity relation analysis.

Table 13: The spatiotemporal understanding QA for duration pattern analysis.

**Task Description**

We are tasked with determining the optimal location for a new electric vehicle (EV) charging station to maximize benefits by effectively meeting demand. The candidate locations, along with their expected charging station coverage, demand (number of vehicles), average distance, travel time, and waiting time to existing charging stations, are provided below.

**Data Schema**

- coverage: The number of locations that will be within the influence radius of the charging station after its construction.
- demand: The number of vehicles requiring charging near the location.
- distance: The average distance that vehicles near the location must travel to reach existing charging stations.
- waiting_time: The average waiting time at nearby existing charging stations. NaN indicates no station nearby and the highest waiting time.
- charging_time: The average charging time at nearby existing charging stations. NaN indicates no station nearby and the highest charging time.

**Data**

location 1:
    - coverage: 3.068
    - demand: 90827.49
    - distance: 188.79m
    - waiting_time: 0.05h
    - charging_time: 22.14h
location 2:
    - coverage: 3.059
    - demand: 105897.62
    - distance: 197.29m
    - waiting_time: 0.03h
    - charging_time: 13.78h
location 3:
    - coverage: 3.059
    - demand: 12443.36
    - distance: 371.80m
    - waiting_time: 0.00h
    - charging_time: 0.96h
location 4:
    - coverage: 3.062
    - demand: 22993.54
    - distance: 304.35m
    - waiting_time: 0.00h
    - charging_time: 1.01h

**Question**

Based on the data provided, rank the locations by average charging time at near charging stations, from the longest to the shortest.

A. 4 > 1 > 2 > 3
B. 3 > 4 > 1 > 2
C. 4 > 1 > 2 > 3
D. 1 > 2 > 4 > 3
**Note**

Please answer the question using the information given. Let's work this out in a step-by-step way to be sure that we have the right answer. Then provide your final answer within the XML tag `<Answer>A/B/C/D</Answer>`.

---

✓ **Qwen2.5-32B**: The locations are ranked by average charging time at nearby charging stations from the longest to the shortest as follows: Location 1 (22.14h), Location 2 (13.78h), Location 4 (1.01h), and Location 3 (0.96h). Option D.

✓ **DeepSeek-R1**: The ranking based on average charging time from longest to shortest is determined by comparing each location's charging_time. Location 1 has the longest at 22.14h, followed by Location 2 (13.78h), then Location 4 (1.01h), and finally Location 3 (0.96h). This results in the order 1 > 2 > 4 > 3, which corresponds to option D. Option D.

✓ **o4-mini**: Location 1 has the longest average charging time (22.14 h), followed by Location 2 (13.78 h), then Location 4 (1.01 h), and finally Location 3 (0.96 h). Option D.

---

Figure 11: The LLMs' reasoning on duration pattern analysis.

Table 14: The spatiotemporal understanding QA for chronological pattern analysis.

**Task Description**

You will analyze a user's historical stay records and predict their visiting patterns.
**Data Schema**

- History: The user's historical travel sequence over the past 30 days.
- Recent: The user's recent visit sequence.
- trajectory: A list of tuples (POI, weekday, time), representing user visits.
- adjacency: A list of tuples (POI_1, POI_2, distance), representing spatial relationships between POIs.
**Data**

History:

Day 1:
- trajectory: [(Train Station 152, Thursday, 08:10 PM)]
- adjacency: N/A

Day 2:
- trajectory: [(Train Station 152, Friday, 10:02 AM), (Building 20014, Friday, 06:01 PM)]
- adjacency: N/A

Day 3:
- trajectory: [(Train Station 152, Monday, 10:01 AM), (Building 20014, Monday, 10:13 AM), (Coffee Shop 14357, Monday, 06:37 PM)]
- adjacency: `[(Building 20014, Train Station 152, 773.0m), (Building 20014, Coffee Shop 14357, 2886.0m)]`
...
Day 30:
- trajectory: [(Train Station 152, Monday, 09:58 AM)]
- adjacency: N/A
**Question**

Based the user's historical stays, identify the most likely pattern in their behaviors:

A. The user usually goes to the Coffee Shop after Building.
B. The user usually goes to the Building after Building.
C. The user usually goes to the Train Station after Train Station.
D. The user usually goes to the Train Station after Building.
**Note**

Please answer the question using the information given. Let's work this out in a step-by-step way to be sure that we have the right answer. Then provide your final answer within the XML tag `<Answer>A/B/C/D</Answer>`.

✓ **Qwen2.5-32B**: The user frequently transitions from Building 20014 to Train Station 152. Option D.

✓ **DeepSeek-R1**: The user frequently transitions from Building 20014 to Train Station 152. This pattern occurs on Days 15, 18, 28, and 29, supported by adjacency data showing these POIs are 773m apart. While other patterns (like same-POI visits) exist, the Building-to-Train Station sequence is the most consistent and spatially logical behavior. Option D.

✓ **o4-mini**: While Train→Train and Building→Train both occur 4 times, the conditional probability of going to a Train Station after a Building visit (4 out of 7) is much higher than going to another Train Station after a Train Station visit (4 out of 14). Option D.

Figure 12: The LLMs' reasoning on chronological pattern analysis.

Table 15: The spatiotemporal understanding QA for trend pattern analysis.

**Task Description**

You are analyzing traffic congestion in a road network of Beijing, China. Given a target road and its nearby neighboring roads, along with time-stamped traffic conditions for each, perform a congestion analysis.

**Data Schema**

- *Target road*: The road for which congestion patterns need to be analyzed.
- *Nearby roads*: Roads within up to two hops from the target road.
- *speed_limit*: Speed limit on the road, with levels from 1 (lowest) to 4 (highest).
- *congestion_level*: A series of congestion levels (from 0 to 4, where 4 is the highest) recorded at specific times during the day.
- *congestion_level_in_past_3_days*: Historical time-series data of congestion levels over the past three days, recorded at specific time intervals.
- *Connectivity*: A list of tuples in the format (road_1, road_2, distance), defining directed connections between roads with the specified distance in meters.

**Data**

*Target road*:
road 33574:
- speed_limit:
- congestion_level: [(4, '17:00'), (4, '17:05'), (4, '17:10'), (4, '17:15'), (4, '17:20'), (4, '17:25'), (4, '17:30'), (3, '17:35'), (3, '17:40'), (3, '17:45'), (2, '17:50'), (3, '17:55')]
- congestion_level_in_past_3_days: [('N/A', '00:00'), ('N/A', '01:00'), ..., (1, '22:00'), (1, '23:00')]
*Nearby roads*:
road 27810:
- speed_limit: 3
- congestion_level: [(1, '17:00'), (1, '17:05'), (1, '17:10'), (1, '17:15'), (1, '17:20'), (1, '17:25'), (1, '17:30'), (1, '17:35'), (1, '17:40'), (1, '17:45'), (1, '17:50'), (1, '17:55')]
...
*Connectivity of roads*: [(22598, 22901, 500m), (27810, 27809, 500m), (33574, 27810, 500m), (33574, 28036, 500m), (28036, 28035, 500m), (22901, 33574, 500m)]

**Question**

Analyze the traffic conditions for target road 33574. What is the overall trend in traffic congestion?
A. No trend
B. Increase
C. Decrease
D. No trend first, then increase
E. No trend first, then decrease
F. Increase first, then no trend
G. Decrease first, then no trend
H. Increase first, then decrease
I. Decrease first, then increase

**Note**

Please answer the question using the information given. Let's work this out in a step-by-step way to be sure that we have the right answer. Then provide your final answer within the XML tag
`<Answer>A/B/C/D/E/F/G/H/I</Answer>`.

> ✗ **Qwen2.5-32B**: The congestion levels for road 33574 show a peak in the late afternoon, followed by a slight decrease. Historical data indicates an increase in congestion levels during the late afternoon and early evening. Nearby roads also show varying levels of congestion, with some experiencing high congestion levels. Option F.
>
> ✓ **DeepSeek-R1**: The target road 33574 shows congestion level 4 (highest) consistently from 17:00 to 17:30, indicating no initial trend. Starting at 17:35, congestion decreases to level 3 and fluctuates slightly (briefly dropping to 2 at 17:50 before rising back to 3 at 17:55). The dominant pattern is a sustained reduction from the initial peak, aligning with option E: 'No trend first, then decrease'.
>
> ✗ **o4-mini**: Congestion on road 33574 is stable at a high level from 17:00–17:30, then shows a clear decreasing trend between 17:30–17:50, followed by a small rebound (increase) at 17:55. Option I.

Figure 13: The LLMs' reasoning on trend pattern analysis.
Table 16: The spatiotemporal understanding QA for local extrema analysis.

**Task Description**

You are analyzing traffic congestion in a road network of Beijing, China. Given a target road and its nearby neighboring roads, along with time-stamped traffic conditions for each, perform a congestion analysis.

**Data Schema**

- Target road: The road for which congestion patterns need to be analyzed.
- Nearby roads: Roads within up to two hops from the target road.
- speed_limit: Speed limit on the road, with levels from 1 (lowest) to 4 (highest).
- congestion_level: A series of congestion levels (from 0 to 4, where 4 is the highest) recorded at specific times during the day.
- congestion_level_in_past_3_days: Historical time-series data of congestion levels over the past three days, recorded at specific time intervals.
- Connectivity: A list of tuples in the format (road_1, road_2, distance), defining directed connections between roads with the specified distance in meters.

**Data**

Target road:
road 27303:
- speed_limit: 4
- congestion_level: [(1, '06:00'), (1, '06:05'), (1, '06:10'), (1, '06:15'), (1, '06:20'), (1, '06:25'), (1, '06:30'), (1, '06:35'), (2, '06:40'), (2, '06:45'), (1, '06:50'), (2, '06:55')]
- congestion_level_in_past_3_days: [(1, '00:00'), (1, '01:00'), (1, '02:00'), (1, '03:00'), (1, '04:00'), (1, '05:00'), (1, '06:00'), (1, '07:00'), (1, '08:00'), (1, '09:00'), (1, '10:00'), (1, '11:00'), (1, '12:00'), (1, '13:00'), (1, '14:00'), (1, '15:00'), (1, '16:00'), (1, '17:00'), (1, '18:00'), (2, '19:00'), (2, '20:00'), (1, '21:00'), (1, '22:00'), (1, '23:00'), (1, '00:00'), (1, '01:00'), (1, '02:00'), (1, '03:00'), (1, '04:00'), (2, '05:00'), (1, '06:00'), (2, '07:00'), (1, '08:00'), (1, '09:00'), (1, '10:00'), (1, '11:00'), (1, '12:00'), (1, '13:00'), (1, '14:00'), (1, '15:00'), (1, '16:00'), (1, '17:00'), (1, '18:00'), (2, '19:00'), (1, '20:00'), (1, '21:00'), (1, '22:00'), (1, '23:00'), (1, '00:00'), (1, '01:00'), (1, '02:00'), (1, '03:00'), (1, '04:00'), (1, '05:00'), (1, '06:00'), (1, '07:00'), (1, '08:00'), (1, '09:00'), (1, '10:00'), (1, '11:00'), (1, '12:00'), (1, '13:00'), (1, '14:00'), (1, '15:00'), (1, '16:00'), (1, '17:00'), (2, '18:00'), (1, '19:00'), (1, '20:00'), (1, '21:00'), (1, '22:00'), (1, '23:00')]
Nearby roads:
road 33889: speed_limit 4, congestion_level: [(1, '06:00'), ..., (2, '06:55')]
...
Connectivity of roads:
`[(23216, 27371, 500m), ..., (27371, 27370, 500m)]`

**Question**

Based on the data, identify one of the typical time periods in the past 3 days when the traffic of the target road 27303 reaches its peak.
A. 18:00-20:00
B. 00:00-02:00
C. 16:00-18:00
D. 06:00-08:00

**Note**

Please answer the question using the information given. Let's work this out in a step-by-step way to be sure that we have the right answer. Then provide your final answer within the XML tag
`<Answer>A/B/C/D</Answer>`.

Figure 14: The LLMs' reasoning on local extrema analysis.
Table 17: The spatiotemporal understanding QA for periodicity pattern analysis.

**Task Description**

You are analyzing traffic congestion in the road network of Beijing, China. Given a target road and its nearby neighboring roads, along with time-stamped traffic conditions for each, perform a congestion analysis.

**Data Schema**

- Target road: The road for which congestion patterns need to be analyzed.
- Nearby roads: Roads within up to two hops from the target road.
- speed_limit: Speed limit on the road, with levels from 1 (lowest) to 4 (highest).
- congestion_level: A series of congestion levels (from 0 to 4, where 4 is the highest) recorded at specific times during the day.
- congestion_level_in_past_3_days: Historical time-series data of congestion levels over the past three days, recorded at specific time intervals.
- Connectivity: A list of tuples in the format (road_1, road_2, distance), defining directed connections between roads with the specified distance in m.

**Data**

Target road:
road 21501:
- speed_limit: 4
- congestion_level: [(2, '22:00'), (2, '22:05'), (2, '22:10'), (2, '22:15'), (1, '22:20'), (1, '22:25'), (1, '22:30'), (2, '22:35'), (2, '22:40'), (1, '22:45'), (1, '22:50'), (1, '22:55')]
- congestion_level_in_past_3_days: [(1, '00:00'), (1, '01:00'), (1, '02:00'), (1, '03:00'), (1, '04:00'), (1, '05:00'), (1, '06:00'), (1, '07:00'), (1, '08:00'), (1, '09:00'), (1, '10:00'), (1, '11:00'), (1, '12:00'), (1, '13:00'), (1, '14:00'), (1, '15:00'), (1, '16:00'), (2, '17:00'), (2, '18:00'), (2, '19:00'), (2, '20:00'), (2, '21:00'), (1, '22:00'), (1, '23:00'), (1, '00:00'), (1, '01:00'), (1, '02:00'), (1, '03:00'), (1, '04:00'), (1, '05:00'), (1, '06:00'), (2, '07:00'), (1, '08:00'), (1, '09:00'), (1, '10:00'), (1, '11:00'), (1, '12:00'), (1, '13:00'), (1, '14:00'), (1, '15:00'), (1, '16:00'), (2, '17:00'), (1, '18:00'), (2, '19:00'), (2, '20:00'), (2, '21:00'), (2, '22:00'), (2, '23:00'), (1, '00:00'), (1, '01:00'), (1, '02:00'), (1, '03:00'), (1, '04:00'), (1, '05:00'), (1, '06:00'), (2, '07:00'), (1, '08:00'), (1, '09:00'), (1, '10:00'), (1, '11:00'), (1, '12:00'), (1, '13:00'), (1, '14:00'), (1, '15:00'), (1, '16:00'), (2, '17:00'), (2, '18:00'), (2, '19:00'), (2, '20:00'), (2, '21:00'), (1, '22:00'), (1, '23:00')]
Nearby roads:
...
Connectivity of roads:
...

**Question**

Based on the data, identify the most possible periodicity of congestion changes on the target road 21501 over the past 3 days.
A. 29 hours
B. 17 hours
C. 24 hours
D. 32 hours

**Note**

Please answer the question using the information given. Let's work this out in a step-by-step way to be sure that we have the right answer. Then provide your final answer within the XML tag `<Answer>A/B/C/D</Answer>`.

> ✓ **Qwen2.5-32B**: The congestion levels on the target road 21501 repeat every 24 hours, indicating a clear 24-hour periodicity. Option C.
>
> ✓ **DeepSeek-R1**: The congestion levels on road 21501 show recurring peaks at 17:00 (5 PM) each day, aligning with typical evening rush hours. Additionally, morning peaks at 07:00 (7 AM) appear on Days 2 and 3. These patterns repeat daily, indicating a 24-hour periodicity corresponding to daily traffic cycles. The 24-hour cycle (Option C) is the most plausible as it matches common diurnal traffic patterns.
>
> ✓ **o4-mini**: The strong evening congestion surge (levels rising to 2) recurs at the same local time each day, indicating a 24 hour cycle. Option C.

Figure 15: The LLMs' reasoning on periodicity pattern analysis.

Table 18: The forecasting QA.

**Task Description**

You are analyzing traffic congestion in a road network of Beijing, China. Given a target road and its nearby neighboring roads, along with time-stamped traffic conditions for each, perform a congestion analysis.

**Data Schema**

- Target road: The road for which congestion patterns need to be analyzed.
- Nearby roads: Roads within up to two hops from the target road.
- speed_limit: Speed limit on the road, with levels from 1 (lowest) to 4 (highest).
- congestion_level: A series of congestion levels (from 0 to 4, where 4 is the highest) recorded at specific times during the day.
- congestion_level_in_past_3_days: Historical time-series data of congestion levels over the past three days, recorded at specific time intervals.
- Connectivity: A list of tuples in the format (road_1, road_2, distance), defining directed connections between roads with the specified distance in m.

**Data**

Target road:
road 28340:
- speed_limit: 2
- congestion_level: [(1, '19:00'), (1, '19:05'), (1, '19:10'), (1, '19:15'), (1, '19:20'), (1, '19:25'), (1, '19:30'), (1, '19:35'), (1, '19:40'), (1, '19:45'), (2, '19:50'), (2, '19:55')]
- congestion_level_in_past_3_days: [(1, '00:00'), (1, '01:00'), (1, '02:00'), (1, '03:00'), (1, '04:00'), (1, '05:00'), (1, '06:00'), (2, '07:00'), (3, '08:00'), (2, '09:00'), (2, '10:00'), (1, '11:00'), (1, '12:00'), (1, '13:00'), (2, '14:00'), (1, '15:00'), (1, '16:00'), (2, '17:00'), (2, '18:00'), (2, '19:00'), (1, '20:00'), (1, '21:00'), (1, '22:00'), (1, '23:00'), (1, '00:00'), (1, '01:00'), (1, '02:00'), (1, '03:00'), (1, '04:00'), (1, '05:00'), (1, '06:00'), (2, '07:00'), (3, '08:00'), (1, '09:00'), (1, '10:00'), (1, '11:00'), (1, '12:00'), (1, '13:00'), (2, '14:00'), (2, '15:00'), (1, '16:00'), (2, '17:00'), (2, '18:00'), (2, '19:00'), (1, '20:00'), (1, '21:00'), (1, '22:00'), (1, '23:00'), (1, '00:00'), (1, '01:00'), (1, '02:00'), (1, '03:00'), (1, '04:00'), (1, '05:00'), (1, '06:00'), (2, '07:00'), (2, '08:00'), (1, '09:00'), (1, '10:00'), (2, '11:00'), (1, '12:00'), (2, '13:00'), (2, '14:00'), (2, '15:00'), (2, '16:00'), (2, '17:00'), (1, '18:00'), (2, '19:00'), (1, '20:00'), (1, '21:00'), (1, '22:00'), (1, '23:00')]
Nearby roads:
road 22668:
- speed_limit: 2
- congestion_level: [(1, '19:00'), (1, '19:05'), (1, '19:10'), (1, '19:15'), (1, '19:20'), (1, '19:25'), (1, '19:30'), (1, '19:35'), (1, '19:40'), (1, '19:45'), (1, '19:50'), (1, '19:55')]
road 27874:
- speed_limit: 2
- congestion_level: [(1, '19:00'), (1, '19:05'), (1, '19:10'), (1, '19:15'), (1, '19:20'), (1, '19:25'), (1, '19:30'), (1, '19:35'), (1, '19:40'), (1, '19:45'), (1, '19:50'), (2, '19:55')]
road 27875:
- speed_limit: 2
- congestion_level: [(1, '19:00'), (1, '19:05'), (1, '19:10'), (1, '19:15'), (1, '19:20'), (1, '19:25'), (1, '19:30'), (1, '19:35'), (1, '19:40'), (1, '19:45'), (1, '19:50'), (2, '19:55')]
Connectivity of roads:
```
[(27874, 28340, 500m), (27875, 27874, 500m), (28340, 22668,
500m)]
```

**Question**

Based on the traffic conditions of the target road and its nearby roads, what is the most possible traffic congestion level of the target road 28340 at the next time step?

A. 1
B. 2
C. 3
D. 4

**Note**

Let's solve this step by step. Finally, summarize your analysis, and provide your answer in JSON format, like:

```
{
  "summary":  "YOUR_SUMMARY",
  "answer":  "A/B/C/D"
}
```

✓ **Qwen2.5-32B**: The current congestion levels of the target road 28340 and its nearby roads are mostly 1, with a slight increase to 2 in the last few minutes. The historical data suggests that the congestion level tends to increase slightly during certain times of the day, but it remains relatively low. Given the current congestion levels and the historical pattern, the most likely congestion level for the next time step is 2. Option B.

✓ **DeepSeek-R1**: The target road 28340's congestion level increased to 2 at 19:50 and remained at 2 at 19:55. Nearby upstream roads 27874 and 27875 also reached congestion level 2 at 19:55, suggesting sustained or increasing traffic inflow. Historical data shows congestion typically drops to 1 at 20:00, but the current trend and upstream conditions indicate the congestion is likely to persist at level 2 in the next time step. Option B.

✓ **o4-mini**: After an initial rise from level 1 to 2 at 19:50–19:55 on road 28340, and seeing the same rise on its upstream neighbor (road 27874), the congestion is likely to remain at level 2 for the next time step. Option B.

Figure 16: The LLMs' reasoning on forecasting.

Table 19: The reflection QA.

**Task Description**

Design a road network to connect unconnected regions to the existing connected regions in an urban area. Using the provided data, identify the roads that can be built to maximize connectivity.

**Data Schema**

- connected_regions: A list of regions that are already connected.
- connectivity: A list of tuples in the format (node_1, node_2, distance), representing undirected roads between nodes with the specified distance (in km).
- unconnected_regions: A list of regions that are currently not connected to any other region.
- region_nodes: A list of boundary nodes defining the extent of each region.
- available_roads: A list of roads that can be built to improve connectivity.
- road_edge: A tuple in the format (node_1, node_2, distance), representing an undirected road that can be constructed between two nodes with the specified distance (in km).
- new_roads: A list of tuples in the format (node_1, node_2, distance), representing new roads that can be constructed after building the previous road.

**Data**

connected_regions:
region 0:
- region_nodes: [3, 2, 1]
region 26:
- region_nodes: [4, 2, 1, 14, 0, 13]
... connectivity:
[(node 1, node 3, 0.8km), (node 0, node 1, 0.17km), ..., (node 44, node 45, 0.36km)]
unconnected_regions:
region 4:
- region_nodes: [7, 9, 17, 5]
region 15:
- region_nodes: [33, 26, 35, 31, 32]
region 16:
- region_nodes: [17, 22, 5, 16, 21]
...
available_roads:

road 7:
- road_edge: (node 3, node 5, 0.31km)
- new_roads: [(node 16, node 5, 0.31km)]
road 20:
- road_edge: (node 17, node 9, 0.14km)
- new_roads: [(node 22, node 17, 0.17km)]
road 30:
- road_edge: (node 15, node 19, 0.52km)
- new_roads: [(node 19, node 25, 0.24km)]
...

**Question**

Based on the provided data, identify the most suitable road should be built next to connect the largest number of unconnected regions.
A. road 7
B. road 72
C. road 84
D. road 81
**Note**

Let's solve this step by step. Finally, summarize your analysis, and provide your answer in JSON format, like:
```
{
  "summary":  "YOUR_SUMMARY",
  "answer":  "A/B/C/D"
}
```

---

✓ **Qwen2.5-32B**: Road 7 can link the largest number of regions (3 regions) and is the best choice to maximize connectivity. Option A.

✓ **DeepSeek-R1**: Road 7 (option A) connects the largest number of unconnected regions (3 regions) according to the analysis. While other roads like 20, 30, 37, 48, and 49 connect 2 regions, road 7's higher initial connectivity impact and the potential to unlock a new road (node 16 to 5) make it the optimal choice for maximizing immediate connectivity gains. Option A.

✓ **o4-mini**: Building road 7 (3 ⟷ 5) immediately connects unconnected regions 4 and 23, and unlocks new_road (16 ⟷ 5) to connect region 16—three regions in total, more than any other candidate. Option A.

Figure 17: The LLMs' reasoning on planning.

Table 20: The spatiotemporal understanding QA for periodicity pattern analysis.

**Task Description**

We are tasked with determining the optimal location for a new electric vehicle (EV) charging station to maximize benefits by effectively meeting demand. The candidate locations, along with their expected charging station coverage, demand (number of vehicles), average distance, travel time, and waiting time to existing charging stations, are provided below.
**Data Schema**

- coverage: The number of locations that will be within the influence radius of the charging station after its construction.
- demand: The number of vehicles requiring charging near the location.
- distance: The average distance that vehicles near the location must travel to reach existing charging stations.
- travel_time: The average travel time for vehicles near the location to reach existing charging stations.
- waiting_time: The average waiting time at nearby existing charging stations. NaN indicates no station nearby and the highest waiting time.
- charging_time: The average charging time at nearby existing charging stations. NaN indicates no station nearby and the highest charging time.
**Data**

location 1:
- coverage: 3.573
- demand: 22734.79
- distance: 284.90m

- waiting_time: 0.00h
- charging_time: 2.38h

location 2:

...

location 3:

...

location 4:

...

**Analysis**

- Coverage ranking:
location 1 (3.573) > location 4 (3.572) > location 3 (3.571) > location 2 (3.571)

- Demand ranking:
location 3 (121545.93) > location 2 (81328.63) > location 4 (27670.77) > location 1 (22734.79)

- Distance ranking:
location 1 (284.90m) > location 4 (158.28m) > location 3 (152.72m) > location 2 (151.30m)

- Waiting time ranking:
location 3 (0.04h) > location 2 (0.04h) > location 4 (0.01h) > location 1 (0.00h)

- Charging time ranking:
location 3 (21.38h) > location 2 (17.03h) > location 4 (8.51h) > location 1 (2.38h)

**Question**

Evaluate each candidate location based on the provided data. Identify which location would deliver the most substantial benefits for an EV charging station, focusing on enhancing overall station coverage.
A. location 1
B. location 2
C. location 3
D. location 4

**Decision**

C

**Reason**

Location 3 offers the highest demand, which is critical for the success and utilization of the EV charging station, making it the most beneficial location despite location 1's higher coverage.

**Environment Feedback**

Station coverage increased by 181.05%
Charging time increased by 7.15%
Travel time increased by 59.22%
Waiting time increased by 83.68%

**Note**

- Based on environment changes caused by the prediction, evaluate the correctness of the prediction and identify any weaknesses.
- Verify if the Decision and its reasoning are accurate.
- If the Decision is incorrect or inaccurate, propose a better alternative with a clear rationale.
- Let's solve this step by step. Finally, summarize your analysis, and provide your answer in JSON format, like:

```
{
  "summary":  "YOUR_SUMMARY",
  "answer":  "A/B/C/D"
}
```

✗ **Qwen2.5-32B**: Location 3 has the highest demand, which is critical for the success of the EV charging station. Despite the high charging time and waiting time, the high demand makes it the most beneficial location. However, improvements in charging time and waiting time are recommended. Option C.

✓ **DeepSeek-R1**: The original decision selected location 3 due to its high demand. However, the environment changes after building there reveal significant increases in waiting time (83.68%), charging time (7.15%), and travel time (59.22%), which undermine the benefits of high demand. While coverage increased substantially (181.05%), the negative impacts on user experience suggest the station may not effectively meet demand. Location 1, with the highest initial coverage, better existing metrics (lowest waiting/charging times), and lower risk of congestion, would provide more balanced benefits without the severe trade-offs observed in location 3. Option A.

✓ **o4-mini**: While Location 3 has the highest demand, it ties for lowest coverage and leads to significant increases in waiting and travel times. Location 1 offers the highest coverage plus zero waiting and the fastest charging, yielding better network performance and user experience. Option A.

Figure 18: The LLMs' reasoning on reflection.

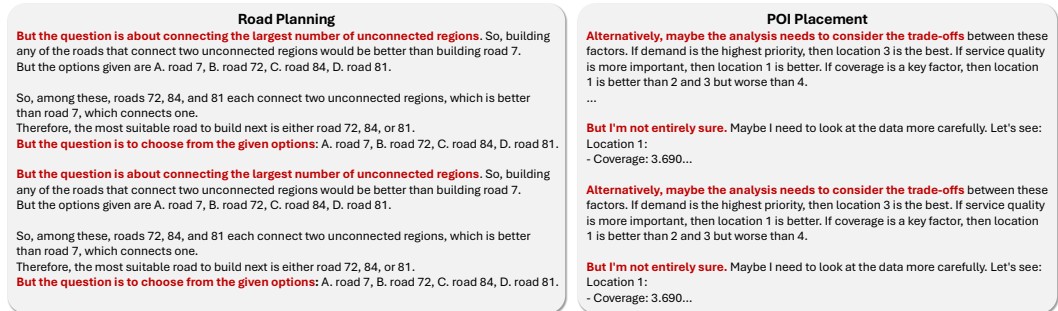

Figure 19: The repetition issues of DeepSeek-R1-Distill-Qwen-7B.

### D.7.4 LLM AGENT PROMPT

Table 21: Task-solving prompt template.

---

**Data Schema**

`<data_schema>`

**Domain Knowledge**

`<domain_knowledge>`

**Data**

`<data_text>`

**Question**

Based on the provided task description, data schema, domain knowledge, provided data, and experience, you may first perform a spatiotemporal data analysis. Then, `<task_target>`.

**Experience**

`<experience>`

**Note**

Let's work this out in a step-by-step way to be sure that we have the right answer. Finally, provide your answer and summarize your analysis in JSON format, like:
```JSON
{
  "answer":  "YOUR_ANSWER",
  "summary":  "YOUR_SUMMARY"
}
```

---

Table 22: Reflection prompt template.

---

**Data Schema**

```
<data_schema>
```

**Domain Knowledge**

```
<domain_knowledge>
```

**Data**

```
<data_text>
```

**Question**

Using the provided task description, data schema, domain knowledge, and data analysis, `<task_target>`.
You need to:
- Evaluate whether the previous `<decision_or_prediction>` was correct, based on the environmental feedback.
- Identify any reasoning flaws or weaknesses in the original output.
- If the `<decision_or_prediction>` was incorrect, propose a better alternative with clear justification.
- Conclude with a high-level lesson learned. This summary must not include specific variable values or data IDs.

**Experience**

```
<experience>
```

**Previous Action**

```
<decision_or_prediction>
```

**Reason**

```
<decision_or_prediction_summary>
```

**Environmental Feedback**

```
<env_feedback>
```

**Note**

Let's work this out in a step-by-step way to ensure we have the right answer. Finally, provide your reflection in JSON format as follows:
```JSON
{
  "is_correct":  "YES/NO",
  "answer":  "YOUR_ANSWER",
  "experience":  "YOUR_SUMMARIZED_EXPERIENCE"
}
```

---

Table 23: Summary prompt template.

---

**Data Schema**

```
<data_schema>
```

**Domain Knowledge**

```
<domain_knowledge>
```

**Data**

```
<data_text>
```

**Question**

Based on the task description, data schema, domain knowledge, and data sample, consolidate all experiences into `<memory_num>` summarized experiences.

**Experience**

```
<old_experience>
<new_experience>
```

**Note**

Let's work this out step-by-step. Combine and restructure the information to create concise and meaningful summaries. Finally, provide your answer in JSON format as follows:
```JSON
[
  "YOUR_EXPERIENCE_1",
  "YOUR_EXPERIENCE_2"
]
```

---

### D.7.5 SPATIOTEMPORAL UNDERSTANDING POST-TRAINING INSTRUCTION EXAMPLES

Table 24: The spatiotemporal understanding post-training instruction for distance analysis.

---

**Task Description**

```
A city is planning to increase green spaces and needs to
consider distances from the new park to residential areas.
```

**Data Schema**

```
- distance:  The distance to the park in km.
```

**Data**

```
neighborhood 0:
- distance:  8.6km
neighborhood 1:
- distance:  8.89km
neighborhood 2:
- distance:  3.86km
neighborhood 3:
- distance:  16.19km
neighborhood 4:
- distance:  6.7km
neighborhood 5:
- distance:  7.96km
neighborhood 6:
- distance:  10.17km
neighborhood 7:
- distance:  13.97km
neighborhood 8:
- distance:  8.39km
neighborhood 9:
- distance:  7.08km
neighborhood 10:
- distance:  6.85km
```

**Question**

How does neighborhood 9 compare to neighborhood 3? Which is closer to the park?

A. neighborhood 9
B. neighborhood 3

**Note**

Let's work this out in a step-by-step way to be sure that we have the right answer. Finally, provide your answer and summarize your analysis in JSON format, like:
```JSON
{
  "answer":  "A/B"
}
```

**Response**
```
<think>
```
Okay, let's see. The question is asking how neighborhood 9 compares to neighborhood 3 in terms of their distance to the new park, and which one is closer. The options are A for neighborhood 9 and B for neighborhood 3.

First, I need to look up the distances for both neighborhoods. From the data provided:
- Neighborhood 3 has a distance of 16.19 km.
- Neighborhood 9 has a distance of 7.08 km.
So, comparing these two numbers, 7.08 km is definitely less than 16.19 km. That means neighborhood 9 is closer to the park than neighborhood 3. Therefore, the answer should be A.
Wait, let me double-check to make sure I didn't mix up the numbers. Neighborhood 3 is over 16 km away, which is quite far, while neighborhood 9 is just over 7 km. Yep, that's correct. The smaller the distance number, the closer they are. So neighborhood 9 is closer. The answer is A.
</think>

```
{
  "answer":  "A"
}
```

Table 25: The spatiotemporal understanding post-training instruction for adjacency analysis.

**Task Description**

```
To enhance pedestrian safety, evaluate the number of crosswalks
near schools and their relationship to traffic volume.
```

**Data Schema**

```
– connectivity:  Each tuple (entity 1, entity 2, distance)
defines a undirected edge from entity 1 to entity 2 with the
specified distance in m.
```

**Data**

```
[(school 1, crosswalk 3, 1623m), (school 1, school 5, 98m),
(school 1, school 4, 1428m), (crosswalk 2, school 4, 1995m)]
```

**Question**

Among school 1 and crosswalk 3, which is further from school 5?

**Note**

Let's work this out in a step-by-step way to be sure that we have the right answer. Finally, summarize your analysis with JSON format, like:
```JSON
{
  "summary":  "YOUR_SUMMARY"
}
```

**Response**
<think>
Okay, let's see. The task is to find which of school 1 or crosswalk 3 has a stronger spatial correlation with school 5. The data given is the connectivity between entities, which includes schools and crosswalks.
First, the definition says that the connectivity is an undirected edge between two entities with a distance. So each tuple represents how those two entities are connected and the distance. I need to look at the data provided and see how school 5 is connected to these entities.
Looking at the data entries: - (school 1, crosswalk 3, 1623m) - (school 1, school 5, 98m) - (school 1, school 4, 1428m) - (crosswalk 2, school 4, 1995m)
School 5 is only mentioned once here, in the second tuple. So school 5 is directly connected to school 1 with a very short distance of 98 meters. That's really close. Now, crosswalk 3 is connected to school 1 but in the same tuple as school 1: (school 1, crosswalk 3, 1623m). So the distance here is 1623 meters, which is much farther.
But how does crosswalk 3 relate to school 5? Since crosswalk 3 is connected to school 1, and school 1 is connected to school 5. So there's an indirect path from crosswalk 3 to school 5 through

school 1. The total distance would be 1623m + 98m = 1721m. Or is it considered as the minimal path? Either way, compared to school 1's direct 98m to school 5, crosswalk 3 is much farther away.

Spatial correlation here likely depends on proximity. The closer two entities are, the stronger their spatial correlation. So since school 1 is extremely close to school 5 (98m), while crosswalk 3 is over 1.6 km away from school 1 and hence over 1.7 km from school 5, school 1 would have a stronger spatial correlation with school 5.

Crosswalk 3 is not directly connected to school 5 in the data. The only connection is through school 1. But even then, the total distance is way larger. The data doesn't have any other connections from crosswalk 3 to other entities except school 1. Therefore, school 1 is directly adjacent and very close, making it the stronger candidate.

Wait, but the problem is about which (school 1 or crosswalk 3) is more correlated spatially with school 5. Since school 1 is directly connected with school 5 at 98m, that's a direct and short link. Crosswalk 3 is connected to school 1 (1623m away) which is connected to school 5 (98m). But crosswalk 3 itself isn't directly linked to school 5. So spatial correlation would depend on distance between entities. School 1 is right next to school 5, making their spatial correlation strong. Crosswalk 3 is much further away. So the answer should be school 1.
```
</think>
{
  "answer":  "school 1",
  "summary":  "School 1 has a stronger spatial correlation with
school 5 because it is directly connected to school 5 via a very
short edge (98 m).  Crosswalk 3 is connected to school 1 but
is much farther away (1623 m) and has no direct connection to
school 5.  Proximity strongly influences spatial correlation,
and school 1's adjacency to school 5 outweighs crosswalk 3's
indirect, distant relationship."
}
```

Table 26: The spatiotemporal understanding post-training instruction for connectivity analysis.

---

**Task Description**

To assess the effectiveness of emergency response, evaluate the accessibility of fire stations to densely populated areas.

**Data Schema**

```
- connectivity:  Each tuple (entity 1, entity 2, distance)
defines an undirected edge from entity 1 to entity 2 with the
specified distance in km.
```

**Data**

```
[(densely populated area 1, densely populated area 7, 5km),
(densely populated area 1, fire station 3, 19km), (densely
populated area 1, densely populated area 4, 6km), (densely
populated area 1, densely populated area 6, 12km), (densely
populated area 1, fire station 2, 4km), (fire station 2, densely
populated area 5, 9km), (fire station 2, fire station 3, 1km),
(fire station 2, densely populated area 6, 12km), (fire station
3, densely populated area 6, 10km), (fire station 3, densely
populated area 4, 14km), (fire station 3, densely populated
area 5, 8km), (fire station 3, densely populated area 7, 9km),
(densely populated area 4, densely populated area 7, 15km),
(densely populated area 5, densely populated area 7, 11km)]
```

**Question**

Is fire station 2 reachable from densely populated area 5?

**Note**

Let's work this out in a step-by-step way to be sure that we have the right answer. Finally, provide your answer and summarize your analysis in JSON format, like:

```JSON
{
  "answer":  "Yes/No",
  "summary":  "YOUR_SUMMARY"
}
```
```
<think>
Okay, let me try to figure out if fire station 2 is reachable from densely populated area 5. I need to see if there's a path connecting them through the given edges. Let's look at the data carefully.
First, let's list out all the connections provided. Each tuple is an undirected edge, so movement can go both ways. The nodes include densely populated areas (like area1, area7, etc.) and fire stations (like fire station 2 and 3).
Starting from densely populated area5. Let's see what connections it has. Looking through the list, I spot (fire station3, densely populated area5, 8km). Oh, right, area5 is connected to fire station3 with a distance. Additionally, there's (fire station2, densely populated area5, 9km). Wait, is that a direct connection between fire station2 and area5? Let me check again. Yes, the tuple (fire station2, densely populated area5, 9km) exists. So if area5 is directly connected to fire station2, then the answer should be yes. But wait, let me confirm if that connection is indeed there.
Looking back at the data provided:
The tuples are as follows:
- (fire station2, densely populated area5, 9km) is one of them. So that means area5 is directly connected to fire station2 with 9km. Therefore, there's a direct path. No need to go through other nodes. Thus, area5 can reach fire station2 directly. So the answer should be Yes.
</think>
{
  "answer":  "Yes",
  "summary":  "Densely populated area 5 is directly connected to fire station 2 via a 9km edge, as per the connectivity data provided."
}
```

Table 27: The spatiotemporal understanding post-training instruction for duration analysis.

**Task Description**

```
Analyze the duration of power outages in industrial zones to
develop strategies for improving grid reliability.
```

**Data Schema**

```
- duration:  The duration of the power outage duration in h.
```

**Data**

```
power outage duration 0:
- duration:  0.1h
power outage duration 1:
- duration:  0.53h
power outage duration 2:
- duration:  1.94h
power outage duration 3:
- duration:  0.82h
power outage duration 4:
- duration:  1.91h
power outage duration 5:
- duration:  1.18h
power outage duration 6:
- duration:  0.8h
```

**Question**

Comparing power outage duration 5 and power outage duration 0, which is longer?

**Note**

Let's work this out in a step-by-step way to be sure that we have the right answer. Finally, provide your answer and summarize your analysis in JSON format, like:
```JSON
{
  "answer":  "power outage duration 5/power outage duration 0"
}
```
<think>

Alright, let's tackle this problem step by step. I need to compare the duration of two specific power outages: number 5 and number 0. The data provided gives the durations in hours, so first I'll list those out.

Looking at the data schema, each entry is labeled as "power outage duration X" with a corresponding duration value. Specifically:

- Power outage duration 0 has a duration of 0.1 hours. - Power outage duration 5 has a duration of 1.18 hours. The question asks which of these two is longer. So, I just need to compare the two numbers: 1.18h vs. 0.1h.

1.18 hours is clearly larger than 0.1 hours. Therefore, power outage duration 5 is longer than duration 0. I can confirm by converting to minutes if necessary. 0.1 hours is 6 minutes, while 1.18 hours is about 70.8 minutes. Yep, that's much longer.

No other data points affect this particular comparison since we are only looking at these two instances. So the answer is straightforward.

</think>

```
{
  "answer":  "power outage duration 5"
}
```

Table 28: The spatiotemporal understanding post-training instruction for chronology analysis.

**Task Description**

You are tasked to analyze a user's historical POI stay records and visiting patterns.

**Data Schema**

- event sequence: Each tuple (event, time, weekday) represents a event that happens at a specific time and day of the week.

**Data**

The POI visiting sequence is below:

Day 1:
- trajectory: [(Bike Shop 4355, Sunday, 06:00 PM), (Convenience Store 2809, Sunday, 07:14 PM)]
- adjacency: N/A

Day 2:
- trajectory: [(Post Office 37, Monday, 09:54 PM), (Bank 8352, Monday, 10:06 PM), (Food and Drink Shop 6831, Monday, 10:06 PM)]
- adjacency: [(Bank 8352, Post Office 37, 467.0m), (Bank 8352, Food and Drink Shop 6831, 117.0m)]

Day 3:
- trajectory: [(Convenience Store 2809, Tuesday, 08:15 PM), (Post Office 37, Tuesday, 08:15 PM), (Food and Drink Shop 1827, Tuesday, 08:15 PM)]
- adjacency: [(Post Office 37, Convenience Store 2809, 731.0m), (Post Office 37, Food and Drink Shop 1827, 471.0m)]

Day 4:
- trajectory: [(Convenience Store 2809, Thursday, 07:50 PM), (Post Office 37, Thursday, 07:58 PM), (Food and Drink Shop 1827, Thursday, 07:59 PM)]
- adjacency: [(Post Office 37, Convenience Store 2809, 731.0m), (Post Office 37, Food and Drink Shop 1827, 471.0m)]

Day 5–30: ...

**Question**

Which event happened most recently?

**Note**

Let's work this out in a step-by-step way to be sure that we have the right answer. Finally, summarize your analysis with JSON format, like:

```
{
```JSON
  "answer":  "Train Station/Park/Convenience Store/Government
Building/Other Great Outdoors/...",
  "summary":  "YOUR_SUMMARY"
}
```
```

**Response**

```
<think>
```

Okay, I need to figure out which event happens most recently based on the data provided. Let's look through all the days and their trajectory entries. The key here is to find the latest time on the latest date.

First, I should check each day's trajectory to see the time and day of the week. All days are numbered from Day 1 to Day 30. Since each day is likely consecutive, Day 30 would be the last one in the sequence. Let me confirm with the data given.

Looking at Day 30:

- The trajectory is [(Train Station 274, Sunday, 05:35 PM), (Miscellaneous Shop 13944, Sunday, 06:02 PM)]

So on Sunday at 06:02 PM, the user visited Miscellaneous Shop 13944.

But wait, are there later days? No, the days go up to Day 30, so that's the latest day in the dataset. Now, looking at the entries on Day 30, the events are at 5:35 PM and 6:02 PM on Sunday.

Are there any other entries in later days with a later time? Let me scan through the other days just to make sure. Let's check Day 29, 28, etc. to see if any times are later in the same day.

Day 29 has a trajectory entry on Saturday at 04:58 AM. That's way earlier. Day 28: Friday, entries up to 07:03 PM. But Saturday would come before Sunday (the next day), so Day 30 is definitely after Day 28 and 29.

Double-check the times on Day 30: the latest time there is 6:02 PM (18:02) on Sunday. Miscellaneous Shop 13944 is the last entry here.

Are there any events on Day 30 that are later than 6:02 PM? The trajectory has two entries: 5:35 PM and 6:02 PM. The latest is 06:02 PM. That's the most recent time in the data.

But wait, maybe another day with a more recent time? For example, maybe a day that's also Sunday but with a later time. Let me check all other Sundays.

Looking through the data:

Day 1 is Sunday (06:00 PM and 07:14 PM). Day 6 is Sunday (04:43 PM). Day 11: Sunday (07:20 PM). Day 16: Sunday (12:29 AM and 12:38 AM). Day 30: Sunday (05:35 PM, 06:02 PM).

The latest Sunday in the dataset is Day 30, and the times there are 5:35 PM and 6:02 PM. But earlier, for example, Day 11 has an event at 7:20 PM. But that Day 11 is earlier (the 11th day), while Day 30 is the 30th day, so even though the time is earlier on a Sunday, the actual date (Day 30) is later. So the Day 30's 6:02 PM is more recent than any entries in previous Sundays, because Day 30 is a later day.

Therefore, the most recent event is Miscellaneous Shop 13944 at 06:02 PM on Sunday. So the answer would be "Miscellaneous Shop".

Wait, in the data entry for Day 30, each event in the trajectory is listed with time. Both entries are on Sunday. The Miscellaneous Shop is the later time. So yes, that's the most recent event in the entire dataset.

```
</think>
{
  "answer":  "Miscellaneous Shop"
}
```

Table 29: The spatiotemporal understanding post-training instruction for trend analysis.

---

**Task Description**

```
You are tasked to analyze the time series data of solar energy
generation power (in MW).
```

**Data Schema**

```
- time-series:  Each tuple (value, time) represents a data
reading in MW recorded at the specified time.
```

**Data**

```
[(30.5MW, 12:00:00), (30.25MW, 12:10:00), (29.8MW, 12:20:00),
(29.7MW, 12:30:00), (29.45MW, 12:40:00), (28.2MW, 12:50:00),
(27.05MW, 13:00:00), (26.65MW, 13:10:00), (26.4MW, 13:20:00),
(25.9MW, 13:30:00), (25.35MW, 13:40:00), (24.6MW, 13:50:00)]
```

**Question**

How is the trend of solar energy generation power (in MW)? First, examine the entire 12-step time series to determine the overall trend. If no clear trend emerges, divide the analysis into two segments of six steps each and assess trends separately for each half.

**Experience**

Let's work this out in a step-by-step way to be sure that we have the right answer.

**Note**

Let's work this out in a step-by-step way to be sure that we have the right answer. Finally, summarize your analysis with JSON format, like:
```JSON
{
  "answer":  "No trend/Increase/Decrease/No trend first, then
increase/No trend first, then decrease/Increase first, then
no trend/Decrease first, then no trend/Increase first, then
decrease/Decrease first, then increase",
  "summary":  "YOUR_SUMMARY"
}
```

**Response**
<think>
Okay, let's tackle this problem step by step. The task is to analyze the time series data of solar energy generation power in MW every 10 minutes from 12:00 to 13:50. The question is asking about the trend—whether it's increasing, decreasing, or if there's no clear trend. If the overall trend isn't clear, we need to split the data into two halves and analyze each separately. First, I'll look at the entire dataset. The timestamps start at 12:00 and go up to 13:50, making 12 data points, each 10 minutes apart. Let's list the values in order: 30.5, 30.25, 29.8, 29.7, 29.45, 28.2, 27.05, 26.65, 26.4, 25.9, 25.35, 24.6 MW. Visually examining these values, each subsequent reading is lower than the previous one. Let's check: From 30.5 to 30.25 (slight decrease), then 29.8, 29.7, 29.45... all the way down to 24.6. Each step shows a reduction. There's a consistent downward trend without any increases. So the overall trend might be a decrease.

But wait, the instruction says to first check the entire 12-step series. Then, if no clear trend, split into two halves. Let me confirm each value again. Starting with 30.5 at 12:00, every next step is lower. For example: Step 1: 30.5 to 30.25 (down) Step 2: 30.5 →30.25 →29.8 →29.7 →29.45 →28.2 (each subsequent is lower). Then from 28.2 to 27.05, down to 24.6. So, all 12 points are steadily decreasing. So the overall trend for the entire period is a decrease.

Wait, no—I might need to check if the rate is consistent. For example, is the decrease steady across all data points? Let's list the values again:

1. 30.5 2. 30.25 (-0.25) 3. 29.8 (-0.45) 4. 29.7 (-0.1) 5. 29.45 (-0.25) 6. 28.2 (-1.25) 7. 27.05 (-1.15) 8. 26.65 (-0.4) 9. 26.4 (-0.25) 10. 25.9 (-0.5) 11. 25.35 (-0.55) 12. 24.6 (-0.75) So the rate of decrease varies. For instance, steps 6 and 7 have larger drops (like -1.25 and -1.15). But even then, every step is lower than the preceding one. So regardless of how much it's decreasing, the trend is consistently downward.

Wait, but the first half (first six steps) shows a downward trend, and the second half (last six steps) also shows a downward trend. So both halves individually have a decreasing trend. That means the entire time series has a consistent decreasing trend. Therefore, the overall answer should be a decrease.

But wait—the problem says if the entire trend is unclear, split into two. But in this case, all points are decreasing. So the trend is clearly a decrease. Hence, the answer should be a single decreasing trend for the entire period. Therefore, the summary would state that the solar generation power shows a consistent decrease across all 12 steps with no periods of increase, leading to an overall downward trend.

```
</think>
{
  "answer":  "Decrease",
  "summary":  "The solar energy generation power shows a
consistent decreasing trend across all 12 time steps.  Starting
at 30.5 MW at 12:00:00, the values decline steadily to 24.6
MW by 13:50:00, with each subsequent measurement being lower
than the previous one.  The rate of decrease varies slightly but
remains unidirectional.  Both the first and second halves of the
data also exhibit decreasing trends, reinforcing the conclusion
of an overall decline."
}
```

Table 30: The spatiotemporal understanding post-training instruction for local extrema analysis.

**Task Description**
You are tasked to analyze the time series data of traffic flow.

**Data Schema**
```
- time-series:  Each tuple (value, time) represents a data
reading recorded at the specified time.
```

**Data**
```
[(36.416666666666664, 2018-02-03 00:00:00), (19.833333333333332,
2018-02-03 01:00:00), (14.083333333333334, 2018-02-03 02:00:00),
(19.333333333333332, 2018-02-03 03:00:00), (36.916666666666664,
2018-02-03 04:00:00), (67.08333333333333, 2018-02-03 05:00:00),
(106.16666666666667, 2018-02-03 06:00:00), (161.75, 2018-02-03
07:00:00)...]
```

**Question**
Identify one of the local maxima peak hours of traffic flow.

**Note**

Let's work this out in a step-by-step way to be sure that we have the right answer. Finally, summarize your analysis with JSON format, like:
```
```JSON {
  "summary":  "YOUR_SUMMARY"
}
```
```

**Response**
```
<think>
```
Okay, let's see. The task is to identify a local maxima peak hour in the traffic flow data provided. So first, I need to understand what a local maximum is in this context. A local maximum in a time series is a point where the value is higher than its immediate neighbors. That means, looking for

times when the traffic flow increased to a peak and then started decreasing. Looking at the data provided, which spans from February 3rd to February 5th, each hour has a traffic value. So I need to go through each hour and check if the current hour's value is higher than the previous and next hour's values. Those would be the local maxima.

Let me list out the values with their times. Starting with February 3rd:

00:00: 36.416..., 01:00: 19.833... → decrease from previous., 02:00: 14.083..., 03:00:19.333..., 04:00:36.916..., 05:00:67.083..., 06:00:106.166..., 07:00:161.75..., 08:00:234.166..., 09:00:256.166..., 10:00:279.83..., 11:00:297.5..., 12:00:309.416..., 13:00:301.25 → This is lower than 12:00's 309.41, so 12:00 is a peak? Let's check. The 12:00 value (309.416) is higher than 11:00 (297.5) and then decreases at 13:00. Yes, so that's a local max. So 2018-02-03 12:00:00 could be a peak.

Then 14:00:285.33, 15:00:276.16, 16:00:306.5 → comparing to 15:00 and 17:00. So 16:00's 306.5 vs previous 276.16 and next hour (17:00) 292.25. So 16:00 on the 3rd is higher than before and after, another local maximum.

Then 17:00:292.25... then it goes down. Then 18th hour: decreases to 255.1, so the peak at 16:00. Looking at Feb 4th. Let me look through their hours. The data from Feb 4th starts at 00:00 again. Let's check peaks there.

For example, at 08:00 Feb 4th: 200.58..., 09:00:239.66, 10:00:239.66, 11:00:272.58, 12:00:280.83, 13:00:283.91, 14:00:300.66. Wait, at 13:00, it's 283.91, then 14:00's 300.66. Is that higher than 15:00? Next hour (15:00 Feb4) is 256.25. So 14:00 is higher than both 13:00 and 15:00; so 14:00 on Feb4 is a local peak.

Then at 19:00 Feb4, the value is 187.91, and 20:00 is 192.33 → not a peak. Let me check other times.

On Feb 5th, the data shows: 05:00:288.41, 06:00:390.08..., 07:00:384.83, 08:00:366.16... So here, 06:00 is higher than 05:00 and 07:00. So that's a peak at 06:00 on Feb5th. Then later, at 17:00 Feb5: 392.16, which is higher than 16:00 (312.25) and 18:00 (305.0). So 17:00 on Feb5 is another local maximum. That's a notable peak.

So possible candidates for local maxima are:
- 2018-02-03 12:00:00 (309.41)
- 2018-02-03 16:00:00 (306.5)
- 2018-02-04 14:00:00 (300.66)
- 2018-02-05 06:00:00 (390.08)
- 2018-02-05 17:00:00 (392.16)

Now, the question asks to identify one of these local maxima. The user just needs one, probably the highest or the most prominent. Let's check the values. The largest numbers I see are from February 5th. The 05:00 on Feb5 is 288, then 06:00 jumps to 390—almost 400, which is a significant spike. Then at 17:00 on Feb5, it goes up to 392.16. Comparing those two, 392.16 is higher than 390.08. Both are peaks. Let me confirm:

05:00 Feb5:288.41 06:00:390.08 (peaks here, since the next hour at 07:00 is 384.83, which is lower) Then at 08:00:366.16 So at 06:00, higher than before and after — peak.

Then 17:00 on Feb5 is 392.16, which follows 16:00 (312.25) and precedes 18:00 (305.0). So that's another peak. So 17:00 is higher than 06:00. Which one should I choose?

The task is to find "one" of them. So selecting either would be correct. However, the highest local maximum would be 17:00 on Feb5 (392.16). Alternatively, the spike in 06:00 on Feb5 may be because it's part of the early morning rush, but the user just wants one example. Maybe the highest one.

But let's verify if there are higher points anywhere else.

Looking at Feb3: max is 309 at 12pm. Feb4's max at 14:00 is 300.6. So Feb5 peaks at 392 and 390.

Yes. So I need to report one. For instance, 2018-02-05 17:00:00 with 392.16 as the local maximum, because that's higher than the surrounding hours. Alternatively, the 06:00 hour on Feb5 is also a local maximum. But since the task allows any one, it's sufficient to pick either.

But the highest is 17:00 on the 5th. Alternatively, perhaps there is a typo, let me check the data again.

Looking at February 5th's 05:00: 288.41. 06:00: 390.08. 07:00: 384.83. So 06:00 is higher than both, so that's a local maximum. Then again, at 17:00, 392.16 is higher than 16:00's 312 and 18:00's 305. So both those times are peaks.

Of these, 392 at 17:00 is higher than the 390 at 6:00. So 17:00 is a higher peak.

So the correct answer would likely be either of these. So to answer the user's question, I can choose the 17:00 on Feb5 as one example.

```
</think>
{
  "answer":  "2018-02-05 17:00:00 identified as a local maximum
peak hour with traffic flow 392.16.",
  "summary":  "A significant local maximum peak occurs on
2018-02-05 at 17:00:00 with a traffic flow of 392.17.  This hour
is preceded by a rise from 312.25 (16:00) and followed by a drop
to 305.0 (18:00), confirming it as a local peak within the time
series."
}
```

Table 31: The spatiotemporal understanding post-training instruction for periodicity analysis.

**Task Description**
You are tasked to analyze the time series data of electricity consumption (in Wh).

**Data Schema**
- time-series: Each tuple (value, time) represents a data reading in Wh recorded at the specified time.

**Data**

The time-series of electricity consumption (in Wh) in every 1 hour is provided below:
```
[(112.0Wh, 2018-03-11 02:00:00), (114.0Wh, 2018-03-11 03:00:00),
(116.0Wh, 2018-03-11 04:00:00), (114.0Wh, 2018-03-11 05:00:00),
(115.0Wh, 2018-03-11 06:00:00), (122.0Wh, 2018-03-11 07:00:00),
(136.0Wh, 2018-03-11 08:00:00)...]
```

**Question**

How long is the period of electricity consumption (in Wh)?

**Note**

Let's work this out in a step-by-step way to be sure that we have the right answer. Finally, summarize your analysis with JSON format, like:
```
```JSON {
  "summary":  "YOUR_SUMMARY"
}
```
```
<think>
Okay, so the task here is to determine the period of the electricity consumption based on the given time series data. Hmm, the term "period" in time series usually refers to the length of a repeating cycle or pattern. Let me think.

Looking at the data, the timestamps start on 2018-03-11 at 02:00:00 and go through each hour until 2018-03-14 at 01:00:00. Let me first check the timestamps to see over how many days this data spans.

Start date: March 11th at 2 AM. The data includes March 11, 12, 13, and part of the 14th (ending at 1 AM on the 14th). So that's roughly three days. Wait, from March 11th 02:00 to March 14th 01:00 is 24h * 3 days minus one hour. So about 71 hours? Let's count. From the first entry to the last entry: 11th 02:00 to 14th 01:00. The total number of data points is for each hour, and the entries seem consecutive with no gaps. Let me count the data points to check.

Looking at the data provided, each entry increments by one hour. Let's see: first entry at 02:00, next at 03:00, up to the last at 01:00 on the 14th. The list starts with (11th 02:00) and ends with (14th 01:00). Since each entry is hourly, the total hours between first and last would be from 02:00 on 3/11 to 01:00 on 3/14. How many hours is that?

From 3/11 02:00 to 3/14 02:00 would be exactly 3 days (72 hours). But since the last entry is 01:00 on 3/14, it's 72 -1 =71 hours. But each entry corresponds to an hour, so number of data points would be 71+1? Wait, no. Let me see. The first entry is at 02:00, next 03:00, etc. So from 02:00 to 24:00 on 3/11 is 22 entries (hours 2-23 inclusive, which is 22 hours?). Wait, no. Wait,

2 AM to 11 PM is 22 hours (since 24-2=22, but with each hour included. Wait, 02:00, 03:00,...23:00 for 3/11. Then next day starts at 00:00. Hmm, the data goes through 3/11, including the next day's entries. Let me recount the days covered. The data starts on 11th, March at 02:00 and ends on March 14th at 01:00. So 11th, 12th, 13th, and part of 14th.

But the main point here isn't the span of the data but looking for the periodicity in the data. For electricity consumption, typically there's a daily pattern—higher during the day, lower at night. Let's check the values to see if they exhibit a daily cycle.

Looking at the data: each day starts with lower values in the early morning, increasing during the day. For example, March 11th, around 10:00 AM (10:00:00), there's a spike to 543Wh, which seems like a huge jump compared to the previous hours. Then on March 12th, at 10:00, it's 526 Wh, also a spike. Same on March 13th: at 10:00, 552 Wh. So every day at around 10 AM, there's a big increase. After that, consumption remains high during the day and starts to drop in the evening. Then at 20:00 (8 PM) each day, it drops to around 359Wh (March 11), 361 (March 12), 357 (March 13), then drops further to around 160-168Wh in the late evening. Then this pattern repeats each day.

So that's a clear daily period, where each day the consumption follows a similar pattern: low at night, peaks in the late morning/afternoon, then decreases again. So the period here is 24 hours—repeating every day.

But wait the question asks "How long is the period of electricity consumption (in Wh)?" So it's looking for the duration of the cycle period, not the data's total duration but the cycle length. Given the daily repeats, the period is 24 hours.

But need to confirm if the data indeed shows daily periodicity. Let me check across the days. Let's take the first three days.

March 11th at 10:00: 543.0 Wh.

March 12th 10:00:526.0.

March13th 10:00:552.0.

Then each day the peak hours of around 10:00-16:00 or so have high values, while nighttime is lower. Therefore, the repeating cycle is daily, which is 24 hours. Thus, the period is 24 hours.

```
</think>
{
  "summary":  "The electricity consumption exhibits a daily
periodicity of 24 hours.  Patterns repeat every day with lower
consumption during nighttime and higher consumption during
daytime, particularly spiking around late morning to afternoon
hours.  This daily cycle is consistent across all provided data
points from March 11th to March 14th."
}
```

