# OpenReview forum: "USTBench: Benchmarking and Dissecting Spatiotemporal Reasoning Capabilities of LLMs as Urban Agents"
_ICLR.cc/2026/Conference — ICLR 2026 Poster_

### Official Review · Reviewer_WawK · 2025-10-17

**Soundness:** 4
**Presentation:** 4
**Contribution:** 4
**Rating:** 8
**Confidence:** 4

**Summary:**

The paper introduces USTBench, a large‐scale benchmark for evaluating the spatio-temporal reasoning abilities of large language models (LLMs) acting as urban agents. USTBench is built on an interactive environment (UAgentEnv) that simulates nine realistic urban prediction and decision-making tasks (e.g., congestion forecasting, traffic-signal control, urban planning). It decomposes reasoning into four processes (understanding, forecasting, planning, and reflection with feedback) and provides 62,466 QA pairs plus standardized end-to-end task metrics. Thirteen state-of-the-art LLMs (generalist and reasoning-enhanced) are benchmarked, revealing strong performance in understanding/forecasting but persistent weaknesses in long-horizon planning and reflection.

**Strengths:**

- First dataset to systematically dissect spatio-temporal reasoning in urban-agent settings and to evaluate a “reflection-with-feedback” capability missing from earlier city-scale benchmarks. Also explicitly compares with STBench, CityBench, CityGPT, and UrbanPlanBench, highlighting gaps such as process-level metrics and long-horizon planning evaluation.
- Covers 9 heterogeneous tasks, 5 data modalities, and ~62k QA pairs, providing both breadth and depth for evaluation.
Benchmarks 13 state-of-the-art LLMs across four reasoning facets, plus ablations on reflection, yielding actionable insights for model developers.
- Demonstrates that general reasoning post-training does not fully transfer to complex urban domains, motivating domain-adaptive techniques.
- Detailed ablation studies and examination of failure cases offer valuable insights into the spatio-temporal reasoning challenges of current LLMs.

**Weaknesses:**

- The study would be strengthened by an explicit modality ablation or corruption experiment to verify which data sources (e.g., socio-economic vs. POI) are most influential in reasoning outcomes.
- The related work section could be strengthened. In particular, the authors may consider discussing the following relevant works, which could help better situate the current contribution within the existing literature:
   - Urban Agent [1, 2, 3]

[1] Wang, Jiawei, et al. "Large language models as urban residents: An llm agent framework for personal mobility generation." Advances in Neural Information Processing Systems 37 (2024): 124547-124574.

[2] Li, Bowen, et al. "LogiCity: Advancing neuro-symbolic ai with abstract urban simulation." Advances in Neural Information Processing Systems 37 (2024): 69840-69864.

[3] Jiang, Yue, et al. "Urbanllm: Autonomous urban activity planning and management with large language models." arXiv preprint arXiv:2406.12360 (2024).

**Questions:**

Mentioned in weaknesses.

---

> ### Author Response · Authors · 2025-11-21
> **Response to Reviewer WawK**
>
> > [W1] **Modality Ablation**: The study would be strengthened by an explicit modality ablation or corruption experiment to verify which data sources (e.g., socio-economic vs. POI) are most influential in reasoning outcomes.
>
> Thank you for the constructive comment. USTBench is carefully designed to **isolate the influence of different modalities** by assigning only the **task-relevant data sources** to each QA type (e.g., socioeconomic prediction QAs use region connectivity and socioeconomic data; mobility prediction QAs use POI and trajectory data). This design ensures that each QA focuses on modality-specific reasoning without introducing redundant or irrelevant signals.
>
> However, we recognize the value of conducting explicit modality ablation or corruption experiments to better understand the influence of different data sources on reasoning outcomes. We will explore this approach in future work to provide deeper insights into **how different modalities contribute to task performance**. Thank you for highlighting this important aspect.
>
> ---
>
> > [W2] **Related Work Enpansion**: The related work section could be strengthened. In particular, the authors may consider discussing the following relevant works, which could help better situate the current contribution within the existing literature.
>
> Thank you for your constructive feedback. We have thoroughly reviewed the suggested papers and revised **Section 6 (Related Work)** to include a discussion of these contributions. In particular, we now emphasize how these works relate to **urban spatiotemporal reasoning and LLM agent design**. We clarify how USTBench **complements and extends prior benchmarks** by offering fine-grained diagnostics, thereby situating our contribution within the existing literature.

---

### Official Review · Reviewer_smgX · 2025-10-25

**Soundness:** 3
**Presentation:** 2
**Contribution:** 3
**Rating:** 4
**Confidence:** 4

**Summary:**

This paper integrates several scenarios related to urban spatio-temporal understanding and their datasets for LLM Agent. Based on this, this paper further splits the process of solving the above scenarios using LLM Agent into four dimensions including  spatiotemporal understanding, forecasting, planning, and reflection, and then designs USTBench to evaluate LLM performance according to each of these four dimensions, as well as to provide an end-to-end overall performance evaluation.

**Strengths:**

1. The idea of splitting the LLM agent into multiple key dimensions and benchmarking them independently proposed in this paper helps to help researchers understand the role played by each dimension in the whole agent autonomous execution process in a more refined way, and can deepen the understanding of the LLM capability boundary.
2. In this paper, a significant amount of work has been invested in completing the processing of raw scenarios and datasets to QA pairs that can be used for benchmarking.
3. This paper gives the review results and analysis of commonly used LLMs, which helps researchers to quickly understand the existing LLMs in the field.

**Weaknesses:**

1. This paper considers LLM agents as spatiotemporal understanding, forecasting, planning and reflection. The statement is not given any reason in this paper so that I find it very confusing. This seems to be a claim forced by the authors to match the benchmark task they designed. Therefore, a reasonable derivation to show that agents used for urban spatio-temporal reasoning tasks include mainly and only these aspects of capabilities is necessary and will complete the logical chain throughout the text.
2. Following the weakness above, Figure 2 also shows a worrying imbalance, with forecasting, planning, and reflection occupying only a relatively small portion of the overall performance radar graph, while the rest is all about the spatial capabilities and temporal capabilities of the LLM. This seems to indicate that the benchmark proposed in this paper does not focus on or is not capable of assessing the aforementioned high-level capabilities of LLM in a holistic manner, but is only a repetitive assessment of the spatio-temporal comprehension capabilities of LLM mainly (such work already exists).
3. In the table in the experimental section, the different models are split by horizontal lines to divide them into groups, but there is no explanation for this, which is equally confusing.

**Questions:**

1. Why are LLM Agent capabilities for urban spatio-temporal understanding limited to four dimensions?
2. How representative is the LLM chosen for this paper? How was the list of models reviewed determined for this paper? There seems to be a limited number of closed-source LLMs here, with a higher number of DeepSeek-R1 variants.
3. How are the lists of LLMs included selected in Figure 4 and Figure 5, especially Qwen2.5-7B vs. Qwen2.5-32B used for comparison in Figure 5?
4. Why are Table 4/Figure 4 and Table 5/Figure 5 so close? This gives the false impression that the tables and figures are related content.
5. With this benchmark, is it possible to quantitatively assess the impact of a single dimension of performance on an end-to-end LLM agent? Or are there deeper and more comprehensive insights beyond reflection to reflect the “DISSECTING” in the title.

---

> ### Author Response · Authors · 2025-11-21
> **Response to Reviewer smgX Part 1 [1/3]**
>
> > [W1] **Reasoning Decomposition**: This paper considers LLM agents for spatiotemporal understanding, forecasting, planning, and reflection. The statement is not given any reason in this paper, so I find it very confusing. This seems to be a claim forced on the authors by the benchmark task they designed. Therefore, a reasonable derivation showing that agents used for urban spatio-temporal reasoning tasks include only these aspects of capability is necessary and will complete the logical chain throughout the text.
>
> Thank you for the insightful comment. We agree that the decomposition requires clearer justification. To clarify, our four-part decomposition (i.e., spatiotemporal understanding, forecasting, planning, and reflection) is not designed post-hoc to match the benchmark. These four abilities represent a **loop of agent-environment interaction**: **understanding** the environment, **forecasting** future states, **planning** actions, and **reflecting** on outcomes to improve future performance. As discussed in **Sections 4.1.2 to 4.1.5**, these abilities are **foundational skills widely recognized** in the literature for LLM agents in general domains to effectively interact with dynamic environments.
>
> The **interactions and mutual benefits** of these components are further supported by recent research on LLM agents:
>
> - **PERIA** [1] models perceive, reason, imagine, and act, where “Perceive” states understanding, “Imagine” functions as forecasting, and “Act” corresponds to planning and execution, closely paralleling our decomposition.
> - **PreAct** [2] demonstrates that forecasting enhances planning, showing that these two processes are operationally distinct and mutually reinforcing.
> - **ReflAct** [3] demonstrates goal-state reflection, where reasoning about current states relative to objectives improves decision quality and reduces error propagation.
>
> While our study establishes a **core**, **interpretable**, and **empirically testable decomposition** to measure LLM agent performance in spatiotemporal reasoning, it does **not attempt to cover all aspects** of LLM agents relevant to broader urban tasks (e.g., social reasoning, multi-agent interaction).
>
> We have revised the manuscript to make this motivation explicit and strengthen the connection to existing LLM agent work. We also acknowledge limitations regarding other reasoning abilities pertinent to broader urban contexts that are not addressed in our study.
>
> ### Reference
>
> [1] Ni, Fei et al. "Peria: Perceive, reason, imagine, act via holistic language and vision planning for manipulation." NeurIPS 2024.
>
> [2] Fu Dayuan et al. "PreAct: Prediction enhances agent’s planning ability." COLING 2025.
>
> [3] Jeonghye Kim et al. "ReflAct: World-Grounded Decision Making in LLM Agents via Goal-State Reflection." EMNLP 2025.
>
> ---
>
> > [W2] **Benchmark Composition**: Figure 2 shows a worrying imbalance, with forecasting, planning, and reflection occupying only a relatively small portion of the overall performance radar graph, while the rest is all about the spatial capabilities and temporal capabilities of the LLM. This seems to indicate that the benchmark proposed in this paper does not focus on or is not capable of assessing the aforementioned high-level capabilities of LLM holistically, but is only a repetitive assessment of the spatio-temporal comprehension capabilities of LLM mainly (such work already exists).
>
> We thank the reviewer for raising this point regarding **Figure 2**. We would like to clarify that the radar chart reflects the **performance profile** of current LLMs, not the **composition of the benchmark** itself. Unlike existing benchmarks, USTBench provides **process-based diagnostics** (**Figure 1**) with a **diverse spectrum** of reasoning abilities, ranging from **foundational** spatiotemporal understanding to **higher-level** forecasting, planning, and reflection. This design enables fine-grained analysis of reasoning failures and error propagation.
>
> As shown in **Table 2**, USTBench explicitly evaluates higher-level capabilities: of the 62,466 structured QA pairs, **Forecasting (15,336), Planning (15,000), and Reflection (8,130),** constructed from **9 representative real-world urban tasks**. These **higher-levels of QAs** together account for approximately **60% of the benchmark**, demonstrating substantial emphasis on these advanced skills. To ensure **comprehensive coverage rather than redundancy**, spatiotemporal understanding itself is decomposed into **8 diverse sub-abilities (Section 4.1.2)**
>
> Moreover, USTBench also includes **end-to-end task evaluations** across **9 real-world urban tasks**, enabling analysis of how understanding, forecasting, planning, and reflection interact within a complete task pipeline.
>
> In summary, USTBench is **not a repetitive test** of spatiotemporal understanding. Instead, it provides a systematic and well-balanced framework that places significant emphasis on **higher-order reasoning**.

---

> > ### Comment · Reviewer_smgX · 2025-11-26
> >
> > Although the authors give a response to W1 here, no substantive changes are given in the article.
> > Similarly, W2 fails to have substantive changes in the text to minimize misunderstandings.
> > I therefore consider these two weaknesses to be unaddressed.

---

> ### Author Response · Authors · 2025-11-21
> **Response to Reviewer smgX Part 2 [2/3]**
>
> > [W3] **Table Layout**: In the table in the experimental section, the different models are split by horizontal lines to divide them into groups, but there is no explanation for this, which is equally confusing.
>
> Thank you for your valuable feedback regarding the table in the experimental section. We apologize for any confusion caused by the horizontal lines dividing the models. As stated in **Section 5.1**, we evaluate both **non-reasoning LLMs** (e.g., GPT-4o) and **reasoning models** (e.g., o4-mini). These splits are intended to **categorize the models into distinct groups**: classic/random baselines, non-reasoning LLMs, and reasoning LLMs. To enhance clarity, we have revised the paper to clearly state these categories. Thank you again for your constructive suggestion.
>
> ---
>
> > [Q1] **Spatiotemporal Understanding Taxonomy**: Why are LLM Agent capabilities for urban spatio-temporal understanding limited to four dimensions?
>
> Thank you for your constructive feedback. To clarify, as outlined in **Section 4.1.2**, our analysis of spatiotemporal understanding encompasses **8 widely recognized types** **(not four)**, specifically: Distance, Adjacency, Connectivity, Duration, Local Extrema, Chronology, Periodicity, and Trend. This classification is based on **established definitions** from prior research, allowing for a systematic assessment of LLMs' ability to accurately extract these features.
>
> ---
>
> > [Q2] **Baseline LLM Selection**: How representative is the LLM chosen for this paper? How was the list of models reviewed determined for this paper? There seems to be a limited number of closed-source LLMs here, with a higher number of DeepSeek-R1 variants.
>
> Thank you for your thoughtful comment. Our model selection was designed to enable a controlled comparison between **base models** (non-reasoning) and their **reasoning variants**, matched by **parameter size and architecture**. For example, we compared Qwen2.5-32B with QwQ-32B, and Llama3.3-70B with DeepSeek-R1-Distill-Llama-70B. This approach allowed us to isolate the effect of reasoning-focused post-training on urban spatiotemporal tasks.
>
> We acknowledge that our benchmark includes a higher number of DeepSeek-R1 variants. This is because **DeepSeek provides** **multiple reasoning distillation variants** that match the **architectures** and **parameter sizes** of widely used base models, enabling direct and fair comparisons between reasoning and non-reasoning models.
>
> Regarding closed-source models, we included leading proprietary LLMs available at the time of our study, such as OpenAI’s **GPT-4o (non-reasoning)** and **o4-mini (reasoning)**, to ensure coverage of **state-of-the-art closed-source** **LLMs**. Overall, our results have provided clear insights into how improvements in LLM reasoning can benefit and fall short in real-world urban tasks. We are committed to expanding our benchmark to include a broader range of both proprietary and open-source models in future releases, and release a **leaderboard** available on HuggingFace.
>
> ---
>
> > [Q3] **Contrast LLM Selection**: How are the lists of LLMs included selected in Figure 4 and Figure 5, especially Qwen2.5-7B vs. Qwen2.5-32B used for comparison in Figure 5?
>
> Thank you for your detailed feedback. We clarify our model selection criteria as follows:
>
> 1. **Figure 4:** Our goal was to examine whether LLMs with stronger spatiotemporal understanding benefit downstream forecasting and planning. To validate this, we compared three models: the **base model** (Qwen2.5-7B), its general **reasoning variant** (DeepSeek-R1-Distill-Qwen-7B), and an **urban-specific fine-tuned model** (Qwen2.5-7B-ST). This selection allowed us to isolate the impact of both general and domain-specific reasoning improvements. The results show that urban-specific instruction tuning leads to significant gains in downstream performance, even surpassing the general reasoning variant.
> 2. **Figure 5 (It is Figure 6 in the updated version):** Here, we aimed to assess the impact of reflective reasoning through ablation studies. As detailed in **Section 5.3**, we selected models with **varying reflection abilities**: DeepSeek-R1 (**high** reflection accuracy, 51.79%), Qwen2.5-32B (**moderate** reflection accuracy, 31.84%), and Qwen2.5-7B (**low** reflection accuracy, 18.99%). This range allowed us to systematically evaluate **how reflection quality affects downstream tasks**. Removing reflection led to performance drops in models with strong reflection abilities, but had inconsistent or even negative effects in models with weaker reflection, indicating that low-quality reflection can introduce noise.
>
> We have added further clarifications on our model selection and experimental settings **in the revised manuscript**. Thank you for helping us improve the clarity of our work.

---

> > ### Comment · Reviewer_smgX · 2025-11-26
> >
> > - [Unaddressed] W3: There are still a lot of horizontal lines in the Reasoning LLMs in Table 5, what does this mean? Does it mean that they are categorized by model size? If so, why are 7B and 9B models separated? Why are Non-Reasoning LLMs not categorized in this way?
> > - Q1: Addressed.
> > - [Unaddressed] Q2: I think such an explanation should be in the article.
> > - Q3: Addressed.

---

> ### Author Response · Authors · 2025-11-21
> **Response to Reviewer smgX Part 3 [3/3]**
>
> > [Q4] **Table/Figure Layout**: Why are Table 4/Figure 4 and Table 5/Figure 5 so close? This gives the false impression that the tables and figures are related content.
>
> Thank you for the constructive comment. We apologize for the confusion caused by the original layout. In the revised manuscript, we have repositioned the figures and tables to reflect their content better and avoid unintended associations.
>
> ---
>
> > [Q5] **Reasoning Ability Evaluation Dissection**: With this benchmark, is it possible to quantitatively assess the impact of a single dimension of performance on an end-to-end LLM agent? Or are there deeper and more comprehensive insights beyond reflection to reflect the “DISSECTING” in the title.
>
> Thank you for the constructive feedback. To clarify, our benchmark already provides **independent quantitative measurements of each reasoning capability**, and we examine how reasoning abilities interact across stages:
>
> - **Spatiotemporal understanding →  forecasting/planning (Section 5.2.2):** As shown in **Table 3**, we observe that models with stronger understanding show consistently better downstream reasoning. As shown in **Figure 4**, we further confirm this causally via **urban-specific instruction tuning**, where improving only spatiotemporal understanding significantly boosts forecasting and planning, demonstrating measurable **inter-stage influence**.
> - **Forecasting →  planning (Section 5.2.2):** As reported in **Table 4**, most LLMs achieve promising forecasting accuracy, yet their planning performance remains substantially lower, especially in tasks requiring alignment with **long-term objectives**. Interestingly, in **long-horizon** forecasting tasks (e.g., traffic-OD prediction), non-reasoning base models sometimes outperform their reasoning-enhanced variants. This disparity highlights the increased complexity of planning and reinforces our claim in **Section 4.1.3** that planning is a higher-order ability, **dependent on and extending beyond forecasting**.
> - **Reflection (Section 5.3):** We compare models with and without reflection-enabled reasoning. Stronger reflective ability leads to clear performance improvements, while weaker models may experience no benefit or even minor degradation. This indicates that **reflection meaningfully contributes to task success**, but its utility depends on the underlying reasoning quality of the agent.
>
> To directly address the reviewer’s question, we further conduct additional **component-level ablations** that individually remove spatiotemporal understanding and forecasting processes from the full pipeline. The findings are as follows:
>
> - **Spatiotemporal Understanding**: Removing this module substantially increases MAPE across most models and tasks, confirming that inaccurate early interpretation **injects systematic errors** into subsequent forecasting and planning.
> - **Forecasting**: Eliminating forecasting notably degrades planning performance in DeepSeek-R1, demonstrating that **high-quality intermediate prediction is essential** for long-term decision-making. While Qwen2.5 7B and 32B show a different trend, in which skip forecasting slightly improves planning. This indicates that **noisy or low-quality intermediate predictions can misguide downstream planning**.
>
> These further analyses enable **full quantitative and causal dissection of all reasoning abilities**, revealing how errors propagate across stages, how components interact, and why specific failure modes arise. Further clarifications are provided in **Section 5.3**. Thank you again for the valuable suggestion.
>
> |                                  | Socio-economic Prediction (MAPE) | Socio-economic Prediction (MAPE) | Socio-economic Prediction (MAPE) | Urban Planning (Service) | Urban Planning (Service) | Urban Planning (Service) |
> | -------------------------------- | ------------------------------------ | ------------------------------------ | ------------------------------------ | ------------------------ | ------------------------ | ------------------------ |
> | Model                            | Qwen2.5-7B                           | Qwen2.5-32B                          | DeepSeek-R1                          | Qwen2.5-7B               | Qwen2.5-32B              | DeepSeek-R1              |
> | W/O Spatiotemporal Understanding | 42.36%                               | 7.04%                                | 6.15%                                | 0.6141                   | 0.5948                   | 0.6550                   |
> | W/O Forecasting                  | -                                    | -                                    | -                                    | **0.6459**               | **0.6356**               | 0.6658                   |
> | Full Pipeline                    | **34.57%**                           | **6.00%**                            | **5.24%**                            | 0.5951                   | 0.6335                   | **0.6858**               |

---

> > ### Comment · Reviewer_smgX · 2025-11-26
> >
> > These are good insights from the benchmark. Thank you for your efforts.
> >
> > However, since the key concerns haven't been effectively addressed in the article, I'm keeping my score for now.

---

> > > ### Author Response · Authors · 2025-11-26
> > > **Response to Reviewer smgX**
> > >
> > > We sincerely thank the reviewer for the careful and thorough review. We are glad to hear that you found our benchmark interesting after our previous clarifications. Below are further point-to-point responses to your concerns.
> > >
> > > > ### [W1] **Reasoning Decomposition**: **Limited substantive changes** are given in the article
> > > >
> > >
> > > We apologize for any lack of clarity in our **previous revision** (**marked in blue**). To summarize the additions provided earlier:
> > >
> > > - In **Section 4.1 (line 189)**, we explained that our four-step reasoning decomposition is based on the **agent–environment interaction loop**: understand → forecast → plan → reflect.
> > > - In **Section 7 (line 535)**, we acknowledged **limitations** regarding other reasoning abilities relevant to broader urban contexts that are not addressed in this study.
> > >
> > > To further enhance clarity, we introduce **additional revisions** (**marked in green**):
> > >
> > > - In **Section 4.1 (line 200)**, we clarify the **advantages** of this decomposition and our QA-style, uniform evaluation, which enables fine-grained assessment and provides a coherent, interpretable, and empirically grounded framework for evaluating LLM agents’ spatiotemporal reasoning.
> > > - In **Section 6 (line 517)**, we describe how our decomposition **aligns with prior studies** and highlight the **interactions and mutual benefits** among the four reasoning abilities.
> > >
> > > ---
> > >
> > > > ### [W2] **Benchmark Composition**: **Fails to have substantive changes** in the article
> > > >
> > >
> > > We first clarify the contents included in our **original submission**:
> > >
> > > - As shown in **Table 2**, USTBench explicitly evaluates higher-level capabilities constructed from **9 representative real-world urban tasks**, including **Forecasting QAs (15,336), Planning QAs (15,000), and Reflection QAs (8,130)**. These **higher-level QAs** together account for approximately **60% of the benchmark**, emphasizing advanced reasoning abilities.
> > > - As stated in **Section 4.1.2**, to ensure **comprehensive coverage** without redundancy, spatiotemporal understanding is further decomposed into **8 diverse sub-abilities**.
> > > - As stated in **Section 4.2**, USTBench includes **end-to-end task evaluations** across the 9 urban tasks, enabling analysis of how understanding, forecasting, planning, and reflection interact within complete task pipelines.
> > >
> > > To further improve clarity, we added **additional revisions** (**marked in green**):
> > >
> > > - In **Section 4.1 (line 195)**, we clarify the **distinction** between **USTBench** and **prior studies**: existing benchmarks primarily focus on fundamental understanding or outcome-based metrics, whereas USTBench provides **process-based diagnostics** covering the full understand–forecast–plan–reflect loop.
> > > - In **Section 4.1 (line 198)**, we clarify the QA reasoning evaluation dataset composition: **40%** target **basic** spatiotemporal understanding, and **60%** target **higher-level** reasoning in **real-world task-solving**.
> > >
> > > ---
> > >
> > > > ### [W3] **Table Layout**: **Horizontal lines** in Table 5
> > > >
> > >
> > > We apologize for the confusion in the original submission. The horizontal lines were used to separate different DeepSeek-R1 distillation variants. Due to the length of their full names (e.g., DeepSeek-R1-Distill-Qwen-7B), we had split them across two rows.
> > >
> > > In the **revised Tables 3–5**, we addressed this issue by using **abbreviated names** (e.g., DeepSeek-R1-Distill-Qwen-7B → DeepSeek-R1-7B) and **removed all unnecessary horizontal lines**. Thank you again for your detailed feedback.
> > >
> > > ---
> > >
> > > > ### [Q2] **Baseline LLM Selection**: Explanation to the **baseline model selection**
> > > >
> > >
> > > To improve clarity, we provide the following **additional revisions** (**marked in green**):
> > >
> > > - In **Section 5.1 (line 321)**, we clarify that our model selection was designed to enable controlled comparisons between **base models** (non-reasoning) and their **reasoning variants**, matched by **parameter size and architecture**, allowing us to **isolate the impact** of reasoning-focused enhancements.
> > > - In **Section 5.1 (line 342)**, we note that **DeepSeek-R1** offers multiple **reasoning distillation variants** with identical architectures and sizes, enabling fair comparisons across model scales.
> > > - In **Section 5.1 (line 343)**, we include GPT-4o and o4-mini as leading closed-source non-reasoning and reasoning LLMs at the time of the study, ensuring coverage of **state-of-the-art closed-source LLMs**.
> > >
> > > ---
> > >
> > > We believe that these additional revisions, together with the content included in our original submission, can fully address your concerns. We sincerely thank you again for your careful review and valuable feedback, which have helped improve the clarity and rigor of our study.

---

> ### Comment · Reviewer_smgX · 2025-11-28
>
> I want to raise my score to 8. But why is there an EDIT button in the page?

---

> > ### Author Response · Authors · 2025-11-28
> > **Response to Reviewer smgX**
> >
> > Thank you very much for your careful review and encouraging comments.
> >
> > Regarding the score update, it might be a system issue on the review platform. Still, we sincerely appreciate your recognition of our work and your willingness to update the rating. Thank you again for your support.

---

### Official Review · Reviewer_bS58 · 2025-11-01

**Soundness:** 3
**Presentation:** 3
**Contribution:** 2
**Rating:** 6
**Confidence:** 3

**Summary:**

The paper proposes USTBench, an evaluation benchmark to systematically measure large language models’ (LLMs) spatiotemporal reasoning abilities in urban environments. Built on an interactive environment called UAgentEnv, the benchmark covers four reasoning facets: spatiotemporal understanding, forecasting, planning, and reflection, and supports both fine-grained (process-based) and end-to-end task evaluation. The authors argue that existing urban LLM benchmarks mostly rely on outcome-based metrics (e.g., traffic efficiency, prediction accuracy) and thus may hide reasoning deficits; USTBench instead makes the intermediate reasoning step explicit and measurable, showing, for example, that reasoning-style LLMs like DeepSeek-R1 do not always outperform strong non-reasoning LLMs such as GPT-4o on urban tasks. The paper also releases datasets (62k+ structured QA pairs), environment configs, and scripts to support reproducibility.

**Strengths:**

1. Timely and well-scoped problem. Urban LLM agents are an emerging but under-evaluated direction. Focusing on spatiotemporal reasoning (not just traffic, not just planning) makes the benchmark conceptually coherent.

2. Process-based evaluation is clearly motivated. The paper illustrates, with concrete cases (e.g. congestion trend vs prediction), that outcome-only metrics can give a misleading ranking of reasoning vs non-reasoning models. This is an actual pain point in current LLM-for-planning work.

3. Comprehensive baselines. They evaluate both non-reasoning and reasoning LLMs, including recent RL-on-reasoning models (DeepSeek-R1, QwQ, GLM-Z1, o4-mini, GPT-4o), which makes the claim “general reasoning does not always transfer to urban tasks” convincing.

**Weaknesses:**

1. Ground-truth construction and label fidelity need more quantification. Many QA instances are generated via interactions with UAgentEnv, but the paper does not report inter-annotator agreement or error bounds for the simulation-derived “optimal” actions. For planning QAs, the “exhaustive search” over horizon H could itself be suboptimal or environment-specific.
2. Evaluation still has a strong API/hardware assumption. The runtime table shows some models become very slow (e.g. DeepSeek-R1 via Alibaba API), which limits the practical use of the benchmark, but the paper doesn’t analyze how this affects comparability across models.
3. Numerous grammatical errors: L107 LLMs excels; L322 Reasoning models achieving...
4. The paper repeatedly claims that “UAgentEnv is an interactive environment for urban agents,” but in fact, the entire environment consists only of static JSON tasks (in QA format), without any state transitions, action feedback, or interactive loops.
5. Tasks are single-step question-answering without any decision–feedback loop.
6. Amap data is not an open API and requires a commercial license.
7. The evaluation of CoT and few-shot results is lacking.

**Questions:**

None

---

> ### Author Response · Authors · 2025-11-21
> **Response to Reviewer bS58 Part 1 [1/2]**
>
> > [W1] **Ground-truth Quantification**: Ground-truth construction and label fidelity need more quantification. Many QA instances are generated via interactions with UAgentEnv, but the paper does not report inter-annotator agreement or error bounds for the simulation-derived “optimal” actions. For planning QAs, the “exhaustive search” over horizon H could itself be suboptimal or environment-specific.
>
> Thank you for the insightful feedback. We clarify that our benchmark uses **real-world urban datasets** for spatiotemporal understanding and forecasting QAs. For **planning and control tasks**, where real-world systems rarely provide observable “optimal” decisions due to stochastic dynamics and delayed feedback, we derive ground-truth actions via a **simulation-driven exhaustive search**.
>
> Regarding label fidelity, as stated in **line 319**, the evaluation outcomes produced by our automated metric are **highly consistent with human judgments**. In **Appendix D.3**, we include a **human evaluation** conducted by **five domain experts** on a randomly sampled set of **250 QAs**. This evaluation assesses both answer correctness and the reasoning process (results shown in **Table 6**). We find strong agreement between expert assessments and our exact-match scoring, which supports the reliability of our ground-truth construction and the robustness of our evaluation protocol.
>
> ---
>
> > [W2] **Cost-Effectiveness Comparison**: Evaluation still has a strong API/hardware assumption. The runtime table shows that some models become very slow (e.g., DeepSeek-R1 via Alibaba API), which limits the practical use of the benchmark, but the paper doesn’t analyze how this affects comparability across models.
>
> We thank the insightful comment. To clarify, we refer the reviewer to **Appendix D.7.1**, where we provide a **cost-effectiveness analysis** comparing model performance relative to resource usage. While reasoning LLMs achieve strong performance, their reasoning processes are often verbose and time-consuming, which can limit their suitability for **real-time deployment** scenarios (e.g., traffic management). These findings highlight research opportunities to develop **lightweight and efficient paradigms** for urban spatiotemporal reasoning, accounting for the number of reasoning tokens used and practical runtime constraints.
>
> Notably, over the past two years, inference speed has improved substantially through advances such as FlashAttention‑2 (**≈2× speedup**) and modern engines like FlashInfer, which **reduce latency by 29–69 % while boosting throughput**. At the same time, API costs have dropped significantly: in June 2025, OpenAI **reduced O3 pricing by 80%** (to \$2 input / \$8 output per 1M tokens), and models like GPT-5 nano now cost as little as \$0.40 per million output tokens. These trends indicate a promising future for deploying LLMs as autonomous agents with lower latency and cost.
>
> ---
>
> > [W3] **Grammatical Errors**: L107 LLMs excels; L322 Reasoning models achieving…
>
> We thank the reviewer for the detailed feedback. We have carefully reviewed the entire manuscript and corrected all identified grammatical errors. These revisions have been incorporated in the **updated version** to ensure clarity and readability. We apologize for any confusion caused by the original errors and appreciate the reviewer’s attention to detail, which has helped improve the overall quality and presentation of our work.

---

> ### Author Response · Authors · 2025-11-21
> **Response to Reviewer bS58 Part 2 [2/2]**
>
> > **Decision-feedback Loop Evaluation**
> >
> > [W4] The paper repeatedly claims that “UAgentEnv is an interactive environment for urban agents,” but in fact, the entire environment consists only of static JSON tasks (in QA format), without any state transitions, action feedback, or interactive loops.
> >
> > [W5] Tasks are single-step question-answering without any decision–feedback loop.
>
> Thank you for the detailed comment. We would like to clarify that our evaluation **does include state transitions and interaction loops**, both in the **reasoning evaluation QA** collection and in the **end-to-end task execution** within UAgentEnv.
>
> - **Reasoning Evaluation QA**: As stated in **Section 4.1.5**, each reflection QA is constructed with a **state transition**: the agent receives (1) its **previous action or prediction**, (2) the **current observation**, and (3) the resulting **environmental feedback**. The agent judges whether its prior output was appropriate and, if not, revises it. Accuracy is measured by whether the agent correctly incorporates feedback to adjust its reasoning. This setup explicitly tests whether the model forms causal links between prior decisions and observed outcomes and uses that feedback to improve subsequent reasoning.
> - **End-to-End Downstream Task Evaluation**: As stated in **Section 4.2**, we evaluate LLMs **end-to-end on nine real-world urban tasks** using the full UAgentEnv agent framework. As illustrated in **Figure 3**, each episode follows a multi-stage decision loop: 1) The agent receives task descriptions, schemas, and domain knowledge; 2) It processes real-time urban **environment state** as the spatiotemporal observation; 3) It performs modular reasoning (**understanding → forecasting → planning**) and outputs an action or prediction; 4) The environment returns **feedback** on the output, which triggers a **reflection step** where errors are diagnosed and experiences are added to memory to guide future decisions. This process forms a **closed decision–feedback loop**, and all prompt templates are provided in **Appendix D.7.4**.
>
> In summary, while the benchmark includes QA-based evaluations, the full UAgentEnv framework supports **interactive, feedback-driven reasoning**, not merely static single-step QA.
>
> ---
>
> > [W6] **Map API Usage**: Amap data is not an open API and requires a commercial license.
>
> Thank you for the constructive feedback. To clarify, **we do not use the Amap API** in our work (See **Appendix B.1**). We have thoroughly reviewed all datasets used in this study, and they are **publicly available** and released under licenses that **explicitly allow academic research use** (e.g., MIT License). Our study is conducted **exclusively for non-commercial, academic purposes**, fully adhering to all data-use terms and licensing conditions.
>
> ---
>
> > [W7] **CoT Evaluation**: The evaluation of CoT and few-shot results is lacking.
>
> We appreciate the reviewer’s valuable comment. To clarify, our reasoning evaluations are conducted using **zero-shot Chain-of-Thought (CoT)** prompting. In addition, we provide extensive **qualitative CoT analyses**, including representative reasoning traces (see **Figure 1**) and error cases (see **Appendix G.7.3**), which reveal common failure modes and motivate the need for domain-adaptive reasoning.
>
> Beyond zero-shot evaluation, we also include **few-shot supervised fine-tuning** on spatiotemporal understanding (**Section 5.2.2**). These experiments show that strengthening spatiotemporal understanding yields substantial downstream gains in forecasting and planning, suggesting promising directions for future few-shot, domain-specific tuning strategies for urban spatiotemporal reasoning.
>
> Finally, we include a **human expert evaluation** (**Appendix** **D.3**) in which five urban science experts assess both answer correctness and **reasoning quality** over 250 QA samples. As shown in **Table 6**, the expert judgments closely align with exact-match metrics, supporting the validity of our CoT-based evaluation.

---

### Official Review · Reviewer_UrGc · 2025-11-01

**Soundness:** 3
**Presentation:** 3
**Contribution:** 3
**Rating:** 6
**Confidence:** 4

**Summary:**

The paper introduces USTBench, a novel benchmark designed to evaluate large language models as urban agents, focusing on their ability to reason about and act within dynamic urban environments. It proposes a comprehensive framework that decomposes urban reasoning into four stages: spatiotemporal understanding, forecasting, planning, and reflection. The benchmark is built around an interactive city simulation (UAgentEnv) based on real-world data and includes 62,466 QA pairs for fine-grained evaluation. The paper shows that while LLMs excel at spatial and temporal understanding, they struggle with long-term planning and reflection, highlighting the need for domain-specific adaptation. By providing detailed insights into each reasoning stage, USTBench aims to advance LLMs' applications in urban systems and other dynamic domains.

**Strengths:**

The paper presents a novel and comprehensive benchmark, USTBench, for evaluating large language models as urban agents, marking an original contribution in the field of AI for urban systems evaluation. The contributions are listed below:
1. It decomposes of urban reasoning into four distinct capabilities (spatiotemporal understanding, forecasting, planning, and reflection), which allows process-based diagnostics instead of only outcome-level evaluation. This modular view introduces a new problem formulation that bridges urban computing and cognitive assessment of LLMs.
2. The benchmark integrates real-world urban datasets, nine realistic urban tasks, and an interactive simulation environment (UAgentEnv). The dataset has over 62,000 structured QA pairs, and is meticulously designed to isolate and measure reasoning sub-skills, enhancing the reliability of evaluation. The experimental section is extensive, comparing multiple state-of-the-art models and providing consistent metrics for each reasoning stage.
3. The paper is well-organized and logically presented. It effectively motivates why decomposing reasoning is critical for diagnosing model weaknesses. Figures and tables are intuitive, and the design of each reasoning subtask is clearly justified.
4. Urban intelligence is an emerging but crucial domain for LLM applications, and the proposed benchmark fills a clear gap by providing standardized, fine-grained evaluation protocols.

**Weaknesses:**

While conceptually strong, the paper has several areas for improvement:
1. Limited coverage of real-world complexity. Although UAgentEnv incorporates multiple tasks, it still simplifies many aspects of urban systems (e.g., policy constraints, multi-agent interactions). The lack of uncertainty modeling may limit generalizability to truly dynamic urban settings.
2. Evaluation breadth. Experiments cover a good range of models, but the paper lacks ablation or fine-tuning studies showing whether models can improve with urban-specific instruction tuning or multimodal inputs. This limits insight into potential solution directions.
3. Reflection dimension underdeveloped. The reflection tasks are intriguing, but the methodology for quantifying reflection accuracy feels underexplained. More qualitative analysis (e.g., error types, improvement dynamics) would clarify why reflection fails and how it could be improved.

**Questions:**

1. Simulation validity: How closely does the “optimal” action derived via simulation match human expert planning in urban contexts? Have the authors conducted any expert validation?
2. Data diversity: Does UAgentEnv include data from multiple cities? If not, how might this bias model generalization?
3. Interpretability: Can the benchmark support interpretability analysis, such as tracing which reasoning stage most strongly correlates with task success, to guide future model improvements?

---

> ### Author Response · Authors · 2025-11-21
> **Response to Reviewer UrGc Part 1 [1/3]**
>
> > [W1]: **Limited coverage of real-world complexity**. Although UAgentEnv incorporates multiple tasks, it still simplifies many aspects of urban systems (e.g., policy constraints, multi-agent interactions). The lack of uncertainty modeling may limit generalizability to truly dynamic urban settings.
>
> We appreciate the reviewer’s insightful comments. To clarify, our evaluations are based on **nine widely studied urban tasks** (see **Section 3.1**), with input modalities faithfully derived from **real-world datasets**.
>
> Prediction tasks directly use **real-world data** within a dynamic urban environment (see **Section 4.1.1**), while decision-making tasks are collected using **well-established urban management platforms** (see **Appendix B.2**). The environments are grounded in **real urban statistics** to ensure realism and reliability, and the management policies are aligned with **regulatory and operational constraints** following prior studies (see **Appendix B.1**).
>
> We acknowledge that USTBench does not yet capture the full complexity of real-world urban systems. As stated in future work (**Section 7**), we plan to extend the benchmark to support real-world urban environments, enabling more comprehensive evaluations under dynamic and uncertain conditions.
>
> ---
>
>
> > [W2]: **Evaluation breadth**. Experiments cover a good range of models, but the paper lacks ablation or fine-tuning studies showing whether models can improve with urban-specific instruction tuning or multimodal inputs. This limits insight into potential solution directions.
>
> > [Q3]: **Interpretability**: Can the benchmark support interpretability analysis, such as tracing which reasoning stage most strongly correlates with task success, to guide future model improvements?
>
> Thank you for the constructive feedback. We would like to clarify that our submission already includes several ablation and fine-tuning studies that examine and explain how LLMs’ reasoning abilities can be improved for urban tasks:
>
> - **Reasoning vs. non-reasoning models (Section 5.2.1):** We compare **non-reasoning LLMs** (e.g., Qwen2.5-32B, Llama3.3-70B) with **reasoning-enhanced variants** (e.g., QwQ-32B, DeepSeek-R1-Distill-Llama-70B). While reasoning LLMs generally outperform their non-reasoning counterparts, the advantage is inconsistent, suggesting that general post-training on logical/mathematical problems does not always transfer to urban spatiotemporal reasoning. This motivates us to explore domain-adaptive methods to improve these abilities.
> - **Spatiotemporal understanding and downstream reasoning (Section 5.2.2):** LLMs with stronger spatiotemporal comprehension tend to perform better on forecasting and planning. We validate this by **urban-specific instruction tuning** a base LLM on spatiotemporal understanding, which significantly improves downstream reasoning performance. This confirms the benefit of improved spatiotemporal understanding for forecasting and planning.
> - **Reflection ablation (Section 5.3):** To assess the effectiveness of reflection, we conduct **ablation studies** comparing performance reflective reasoning. Models with strong reflection show clear gains, whereas models with limited reflection may see no benefit or even slight degradation, highlighting that reflection quality matters for urban tasks.
> - **Modality-specific design:** USTBench is carefully designed to **isolate the influence of different modalities** by assigning only the **task-relevant data sources** to each QA type (e.g., socioeconomic prediction QAs use region connectivity and socioeconomic data; mobility prediction QAs use POI and trajectory data). This design ensures that each QA focuses on modality-specific reasoning without introducing redundant or irrelevant signals.
> - **Interpretability**: We provide interpretability-oriented analyses through case studies. **Figure 1** shows that general reasoning LLMs sometimes underperform non-reasoning models, motivating us to explore domain-adaptive methods for urban spatiotemporal reasoning. We also provide a range of example outputs from different LLMs in **Appendix G.4.3**, which illustrate varied failure modes across different tasks. These examples help identify LLMs’ flaws in spatiotemporal reasoning and guide future improvements.
>
> We believe our results already provide clear guidance on how improvements in reasoning propagate to real-world urban tasks. In future work, we plan to explore additional **domain-specific approaches** (e.g., agentic RL [1] with process-based rewards tailored to urban tasks) to further enhance LLMs' spatiotemporal reasoning capabilities as urban agents.

---

> ### Author Response · Authors · 2025-11-21
> **Response to Reviewer UrGc Part 2 [2/3]**
>
> > [W3]: **Reflection dimension underdeveloped**. The reflection tasks are intriguing, but the methodology for quantifying reflection accuracy feels underexplained. More qualitative analysis (e.g., error types, improvement dynamics) would clarify why reflection fails and how it could be improved.
>
> Thank you for the constructive suggestion. We agree that clarifying **how reflection accuracy is quantified** is important. In **Appendix D.7.3**, we have provided representative reflection outputs from different LLMs, illustrating diverse failure modes across urban tasks.
>
> To address the reviewer’s comment, we added a **qualitative study analyzing reflection errors, reasoning quality, and faithfulness**, evaluated using GPT-5 as an LLM-based judge. We consider three models with varying reflection abilities: **Qwen2.5-7B, Qwen2.5-32B, and DeepSeek-R1**. Reflection performance is categorized along three dimensions:
>
> - **Error Types**: 1) *Feedback Interpretation Error*: the model misinterprets the environment feedback. 2) *Adaptation Error*: The model fails to adjust previous outputs appropriately. 3) *Feedback Integration Error*: The model incorporates feedback inconsistently or partially.
> - **Reasoning Quality**: 1) *No Error Identification*: the model fails to detect any errors in its outputs. 2) *Partial Error Identification*: The model identifies some but not all errors. 3) *Correct Fix*: The model successfully identifies and corrects errors. 4) *Incorrect Fix*: The model misidentifies errors or applies incorrect corrections.
> - **Faithfulness**: 1) *Overconfident Wrong*: the model produces highly confident but incorrect reflections. 2) *Inconsistent Reasoning*: The reflection contradicts prior outputs or observations.
>
> We analyzed three models with varying reflection capabilities (Qwen2.5-7B, Qwen2.5-32B, DeepSeek-R1). Key observations include:
>
> - **Error Types:** Stronger models like DeepSeek-R1 show fewer feedback interpretation and adaptation errors than Qwen2.5-7B/32B, yet challenges remain in dynamically integrating feedback, highlighting adaptability limits in complex urban tasks.
> - **Reasoning Quality:** DeepSeek-R1 achieves a higher correct-fix rate, demonstrating superior error detection and correction, whereas Qwen2.5-7B often fails to detect errors, and Qwen2.5-32B shows partial correction ability.
> - **Faithfulness:** Weaker models tend to be overconfident. While DeepSeek-R1 mitigates overconfidence, occasional inconsistent reasoning persists, indicating that faithfulness is not fully resolved by stronger reasoning alone.
>
> In summary, **enhanced reasoning improves error correction and reduces overconfidence but does not fully eliminate inconsistencies**, highlighting important directions for future improvements in reflection and robustness. The corresponding **results and analysis** are included in **Section 5.2.3 (Figure 5)**. We believe this detailed qualitative study directly addresses your feedback and strengthens our work.
>
> | Error Type                        | Qwen2.5-7B | Qwen2.5-32B | Deepseek-R1 |
> | --------------------------------- | ---------- | ----------- | ----------- |
> | **Feedback Interpretation Error** | 33%        | 44%         | 19%         |
> | **Adaptation Error**              | 61%        | 53%         | 34%         |
> | **Feedback Integration Error**    | 6%         | 3%          | 41%         |
> | **Causal Reasoning Error**        | 0%          | 0%           | 6%          |
>
> | Reasoning Quality                | Qwen2.5-7B | Qwen2.5-32B | Deepseek-R1 |
> | -------------------------------- | ---------- | ----------- | ----------- |
> | **No Error Identification**      | 95%        | 99%         | 54%         |
> | **Partial Error Identification** | 1%         | 0%          | 0%           |
> | **Correct Fix**                  | 1%         | 1%          | 46%         |
> | **Incorrect Fix**                | 3%         | 0%          | 0%           |
>
> | Faithfulness               | Qwen2.5-7B | Qwen2.5-32B | Deepseek-R1 |
> | -------------------------- | ---------- | ----------- | ----------- |
> | **Overconfident Wrong**    | 78%        | 96%         | 58%         |
> | **Inconsistent Reasoning** | 22%        | 4%          | 42%         |

---

> ### Author Response · Authors · 2025-11-21
> **Response to Reviewer UrGc Part 3 [3/3]**
>
> > [Q1] **Human Evaluation**: How closely does the “optimal” action derived via simulation match human expert planning in urban contexts? Have the authors conducted any expert validation?
>
> Thank you for the insightful feedback. As stated at **line 347 (Section 5.2.1)**, our evaluation results have been demonstrated to be **consistent with the human evaluation**. In **Appendix D.3**, we conduct a **human evaluation** with **five human experts** in urban science on a randomly selected subset of 250 QAs from our benchmark. This assessment examines not only the correctness of model answers but also evaluates the underlying reasoning process. The results are shown in **Table 6**. We find that human evaluations generally align with our exact-match assessments, supporting the reliability of our automated metric.
>
> ---
>
> > [Q2] **City Coverage**: Does UAgentEnv include data from multiple cities? If not, how might this bias model generalization?
>
> We appreciate the reviewer’s insightful comments. We kindly invite the reviewer to refer to **Section 3.1** and **Appendix B.1**. To clarify, our evaluation currently incorporates urban data from eight geographically and structurally diverse cities: **Beijing, Guangzhou, Jinan, Hangzhou** (China), **New York City** (USA), **Cape Town** (South Africa), **Harare** (Zimbabwe), and **Mumbai** (India). These cities were selected to capture a range of urban dynamics, including high-density megacities, mixed-use transit systems, and diverse socioeconomic profiles.

---

> > ### Comment · Reviewer_UrGc · 2025-11-26
> >
> > Thanks for responding for all the questions. The ablation study, quantification of reflection accuracy, and human evaluation are well-performed, which will definetely make the paper more solid.

---

> > > ### Author Response · Authors · 2025-11-26
> > > **Response to Reviewer UrGc**
> > >
> > > We sincerely appreciate the reviewer’s dedicated and constructive feedback. We are glad that the additional clarifications and details provided in our rebuttal helped address the earlier concerns. Your comments are highly encouraging and help further strengthen our paper.

---

### Author Response · Authors · 2025-11-30
**Official Comment to Program Chair and Area Chair**

**Dear Program Chair and Area Chair,**

We sincerely appreciate your oversight throughout the review process and the reviewers’ constructive feedback, which has improved the clarity, rigor, and presentation of our work. Below, we summarize each reviewer’s major concerns, our responses, and their follow-up remarks. We believe all concerns have been fully addressed in either the **original submission or the revised main text**.

---

**Reviewer UrGc (Rating: 6 → Positive follow-up)**:

- **Real-world complexity**: The reviewer noted potential simplifications. We clarified that our study is grounded in **real-world statistics** and incorporates **regulatory constraints** across **nine urban tasks**, and planned future real-world deployment evaluations.
- **Evaluation breadth**: The reviewer requested ablation/fine-tuning analysis with insights for future improvement. We highlighted that our **ablation, fine-tuning, interpretability, and human evaluation** under modality-isolated settings were already included in the **original submission**.
- **Reflection assessment**: The reviewer requested qualitative analysis of reflection reasoning. We expanded **qualitative analyses** covering error types, reasoning quality, and faithfulness in the **revised main text**.
- **Simulation validity & diversity**: The reviewer questioned expert validation and city diversity. We clarified **human validation** in the **original submission** and coverage of **eight cities** worldwide.

**Follow-up**: After the **initial rebuttal**, the reviewer stated that our additions **“will definitely make the paper more solid,”** indicating **full satisfaction**.

---

**Reviewer bS58 (Rating: 6):**

- **Ground-truth fidelity**: The reviewer requested label quality quantification. We clarified that annotations were verified by **five experts** in the **original submission**.
- **API/hardware usage**: Concerns about latency and cost trade-offs were addressed with a **cost-effectiveness analysis (original submission)**, and we discussed recent advances in **speedups and cost reductions** for deployment of LLM-based urban agents.
- **Decision–feedback loop**: The reviewer questions whether our evaluations incorporate decision-feedback loops. We clarify that **state transitions and feedback** are included in both **process-based QA and end-to-end evaluations**.
- **Evaluation design:** The reviewer requests CoT/few-shot evaluation. We clarified our use of **zero-shot CoT** prompting evaluation and **few-shot fine-tuning** ablation, alongside **human expert evaluation** of both answer correctness and reasoning quality in the **original submission**.
- **Grammer & API Usage**: All grammer errors were corrected in the **revised paper**, and dataset licensing is confirmed **public and available for research**.

---

**Reviewer smgX (Rating: 4 → Expressed intent to raise to 8):**

- **Reasoning decomposition**: The reviewer requested justification of the decomposition of reasoning abilities. We clarified its grounding in the **agent–environment interaction loop** and alignment with **recent LLM agent studies** in the **revised main text**.
- **Benchmark balance**: The reviewer raises concerns about the balance between basic and higher-level reasoning. We clarified that **60%** of QAs target **higher-level reasoning**, while understanding fundamental understanding QAs are organized into **8 sub-abilities** to ensure coverage.
- **Presentation issues**: Tables and figures were **reorganized**, and **baseline model selection** criteria justification was expanded.
- **Dissection of reasoning abilities**: The reviewer requests a quantitative assessment of each reasoning ability. While our **original submission** already included isolated quantitative measurements in **process-based evaluation**, we expanded **component-level dissection** in the end-to-end urban task evaluation in the **revised main text**.

**Follow-up**: After the **initial rebuttal**, the reviewer's concerns remains only on **the reflection of clarifications in the revised paper**. In the end, the reviewer stated an intention to **raise the score to 8**, indicating full resolution.

---

**Reviewer WawK (Rating: 8):**

- **Modality ablation**: The reviewer suggested modality-controlled experiments. We clarified that our design **already isolates modality effects** and outlined plans for **modality-ablation and corruption studies**.
- **Related work**: We expanded the **related work** to incorporate suggested literature.

---

In summary, **two reviewers (UrGc and smgX)** provided **strong positive follow-up**, with smgX intending to **raise the score to 8**. We respectfully request that the AC/PC consider the positive review trajectory and the substantial revisions made. We are confident that USTBench provides a rigorous, timely foundation for advancing spatiotemporal reasoning in LLM-based urban agents.

Thank you for your consideration.

Best regards,

The authors of submission 16983

---

### Meta-Review · Area_Chair_MHS3 · 2026-01-05

**Summary:**

This submission introduces USTBench, a large-scale benchmark for evaluating LLMs as urban agents, with a process-based decomposition into spatiotemporal understanding, forecasting, planning, and reflection, built around an interactive simulation environment (UAgentEnv) and 62k+ QA pairs.

Across reviews, the core consensus is that the benchmark is timely and valuable, with strong design motivation and extensive evaluation. Reviewers particularly appreciate: (i) the process-based diagnostic framing beyond outcome-only metrics, (ii) breadth across 9 urban tasks, and (iii) empirical findings that general “reasoning” post-training does not reliably transfer to urban spatiotemporal reasoning, especially for long-horizon planning and reflection.

After the rebuttal and ensuing discussion, two initially skeptical reviewers (especially smgX) indicate improved confidence and willingness to raise the score substantially (to 8),  implying that the core scientific concerns were addressed. Overall, there will be a consensus of acceptance among the reviewers as rebuttal, thus acceptance is recommended.

**Reviewer Concerns:**

Most of the concerns have been addressed.

**Reviewer Scores:**

Reviewer smgX (initial: 4, borderline reject): smgX was initially the outlier (score 4) and repeatedly said key issues were unaddressed. After the authors’ final point-by-point response, smgX wrote: “I want to raise my score to 8.” So Likely score change: 4 → 8.

So even the other review scores remain the same, there are 8,8,6,6, a consensus of acceptance.

---

### Decision · Program_Chairs · 2026-01-26

Accept (Poster)